# MMReD: a Cross-Modal Benchmark for Dense Context Reasoning

**Maxim Kurkin**[*,1,2], **Boris Shirokikh**[*,1],
**Irina Abdullaeva**[1,3], **Viktoriia Chekalina**[1,4] & **Andrey Kuznetsov**[1,3]
[1]FusionBrain Lab, AXXX, Moscow, Russia
[2]Applied AI Institute, Moscow, Russia
[3]Research Center of the Artificial Intelligence Institute, Innopolis University, Innopolis, Russia
[4]Lomonosov Moscow State University, Moscow, Russia

## Abstract

Despite recent advancements in extending context windows of large language models (LLMs) and large vision-language models (LVLMs), their ability to perform complex multi-modal reasoning over extended contexts remains critically limited. To underline this challenge, we present **MMReD**, a benchmark specifically designed to assess reasoning abilities within dense, information-rich scenarios where simple retrieval is not enough. Unlike traditional Needle-in-a-Haystack evaluations, MMReD challenges models to identify and interpret global patterns across entire contexts. Our benchmark comprises 24 tasks of varying complexity, ranging from standard passkey retrieval setups to those requiring selective or uniform attention to all context chunks. The evaluation reveals a consistent performance drop across all tested models – including the most advanced LLMs, LVLMs, and architectures specializing in code and reasoning – as the number of observations increases. Notably, even the leading reasoning-specialized models achieve 0% accuracy on certain tasks at the maximum context length of 128 observations. Conventional fine-tuning techniques, such as SFT and GRPO, also fail to generalize effectively to longer contexts. These observations reveal an inherent limitation in current model architectures, emphasizing the need for innovative approaches to enable competent dense context reasoning in multi-modal AI systems.

## 1 Introduction

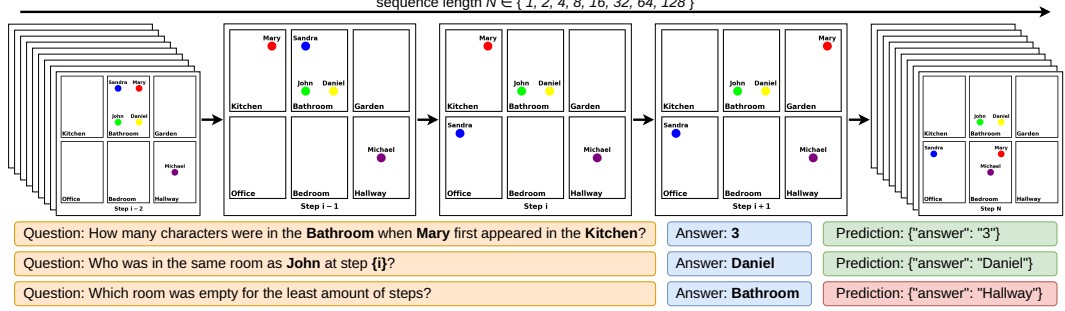

Figure 1: Overview of a multi-modal dense-sequence reasoning task from MMReD. Models are evaluated on queries that require tracking entities, spatial relationships, basic counting, and event-based reasoning over varying sequence lengths $N$, evaluating the models' ability to process and retain long-term dependencies. Example questions cover all designed answer types: numbers, characters, and rooms. In the images and questions, we replace $i$, $i+1$, etc, with the real step numbers.

---

[*]Equal contribution. Correspondence to: `kurkin@fusionbrainlab.com`

Large language models (LLMs) have demonstrated a remarkable ability to reason in both short and long contexts. Long-context extension, in particular, has gained significant attention as models are increasingly deployed in tasks requiring memory, retrieval, and sequential reasoning.

Standard practice on reasoning evaluation is to rely on a range of public benchmarks that test reasoning capabilities on a variety of topics. Short-context reasoning includes general language understanding and reading comprehension (DROP (Dua et al., 2019), ARC (Clark et al., 2018), BBH (Suzgun et al., 2022)), academic reasoning (MMLU-PRO (Wang et al., 2024d), GPQA (Rein et al., 2024), MATH (Hendrycks et al., 2021), AIME (MAA, 2024)) and code comprehension (HumanEval (Chen et al., 2021), CRUX (Gu et al., 2024), MBPP (Austin et al., 2021)). Long-context reasoning includes recent benchmarks RULER (Hsieh et al., 2024), BABILong (Kuratov et al., 2024), and Michelangelo (Vodrahalli et al., 2024).

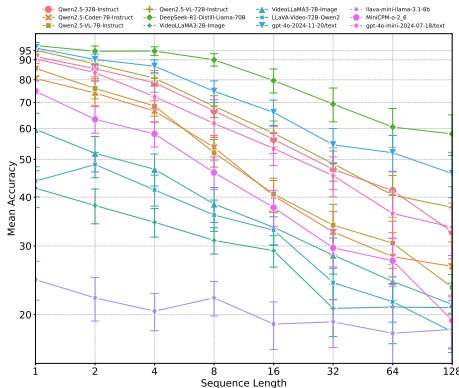

Figure 2: All evaluated LLMs and LVLMs share a common performance profile, decreasing with the context length of MMReD tasks. The results are grouped and averaged across question types.

Large vision-language models (LVLMs) are also rapidly closing the gap in performance on various visual recognition tasks that require reasoning in both text and visual modalities. Existing benchmarks can be categorized by modality of context in the question: single image understanding (MMMU (Yue et al., 2024a;b)), multiple image understanding (MuirBench (Wang et al., 2025), BLINK (Fu et al., 2024b)), and video understanding (VideoMME (Fu et al., 2024a), MLVU (Zhou et al., 2024), EgoSchema (Mangalam et al., 2023), LVBench (Wang et al., 2024b)).

While these benchmarks demonstrate improved performance on longer contexts, they primarily focus, as we show below, on retrieval-based or Needle-in-a-Haystack (NIAH) setups, where models locate a specific fact from an otherwise irrelevant or distractor-filled context. However, such tasks do not fully capture a model's ability to reason across densely distributed information. In our analysis, we show that state-of-the-art LLMs and LVLMs exhibit no clear correlation between their NIAH performance and their ability to perform deeper, structured reasoning in the information-rich scenarios. So NIAH performance on existing benchmarks alone is not a reliable indicator of reasoning ability.

To bridge this gap, we introduce MMReD (Multi-Modal REasoning in Dense context), a benchmark designed to assess a conceptually different capability: reasoning in dense environments where models must attend uniformly across the entire context. Our experiments reveal a consistent decline in performance across all tested models as context length increases (Figure 2), highlighting core limitations in current architectures and training approaches for dense context multi-modal reasoning.

Our contributions can be summarized as follows:

1. We develop and release MMReD, a comprehensive benchmark for evaluating long-context multi-modal reasoning, which goes beyond existing NIAH setups.

2. We demonstrate that state-of-the-art LLMs, LVLMs, and reasoning-specialized architectures fail to generalize to dense context reasoning, revealing novel significant limitations.

3. We show that standard fine-tuning methods, e.g., supervised fine-tuning (SFT) and GRPO (Shao et al., 2024), are insufficient for enabling dense context reasoning in current models.

4. We highlight key challenges and propose potential directions for future research aimed at overcoming the current limitations in dense context reasoning for multi-modal AI systems.

## 2 RELATED WORK

Long-context understanding has become a central challenge in the development of LLMs and LVLMs. Prior research in this area spans (i) benchmarks for long-context comprehension and (ii) architectural and training approaches for improving long-context reasoning. We briefly review both strands, motivating the unique role of MMReD.

**Early efforts in long-context evaluation.** Initial work focused on synthetic or textual setups designed to probe whether models can retain and exploit information over extended sequences. For example, Kim & Schuster (2023) introduced entity-tracking and disambiguation tasks, providing some of the first systematic evidence that transformer models struggle when the relevant signal is deeply buried in the input.

**Rise of the Needle-in-a-Haystack paradigm.** Building on these insights, subsequent benchmarks formalized the retrieval challenge by embedding a small "needle" of task-relevant information inside a large, mostly irrelevant context. BABILong (Kuratov et al., 2024) extended the bAbI reasoning tasks (Weston et al., 2016) into book-length contexts, while Visual Haystacks (Wu et al., 2024) adapted this idea to multimodal settings by injecting known objects into distractor-heavy image collections. These benchmarks popularized the NIAH paradigm, which has since become the dominant template for evaluating long-context models.

**Beyond single-needle retrieval.** Recent work has sought to move past simple NIAH formulations. Vodrahalli et al. (2024) introduced the Michelangelo benchmark to study the "short-circuiting" problem, where models exploit superficial correlations rather than genuinely using the entire context; its most demanding *latent list* setting requires tracking up to 20 relevant needles among distractors, yet state-of-the-art models still perform well above chance. In parallel, Bai et al. (2025b) proposed LongBench v2, which grounds evaluation in naturally occurring long contexts rather than synthetic haystacks, thereby improving ecological validity. Nevertheless, both Michelangelo and LongBench v2 remain closer to multi-needle retrieval problems: the bulk of the context is still dominated by irrelevant or weakly informative content. By contrast, MMReD is explicitly designed so that *all* context elements are densely informative, forcing models to integrate global patterns rather than locate sparse signals.

**Architectural approaches and theoretical limitations.** Research on long-context modeling has explored architectural innovations and training schemes: memory-augmented transformers (Bulatov et al., 2022; Rodkin et al., 2024), structured state-space models such as Mamba (Gu & Dao, 2024; Dao & Gu, 2024), and context-extension techniques like YaRN (Peng et al., 2023) and LongVA (Zhang et al., 2024a). While these approaches extend sequence length handling, their evaluation is typically tied to NIAH-style retrieval. More recently, Litman & Guo (2026) reframe attention through Entropic Optimal Transport, replacing the implicit uniform prior of standard softmax with a learnable continuous prior; beyond providing an EOT-based explanation of attention sinks, this approach yields a prior that combines the flexibility of learned positional embeddings with the length-generalization properties of fixed encodings. Two concurrent theoretical analyses illuminate *why* such failures arise structurally. Veličković et al. (2025) prove that any fixed-temperature softmax-based circuit must *disperse*—attention weights necessarily spread over irrelevant keys as the number of items grows—making sharp, reliable lookups at unseen sequence lengths provably impossible within standard softmax architectures. Ebrahimi et al. (2026) show that transformers learn *length-specific* solutions with negligible weight sharing across sequence lengths, so the data required to cover new lengths grows far more steeply than for recurrent models that amortize learning across lengths; this directly explains why standard SFT and GRPO fail to generalize to longer contexts in our experiments. Taken together, these results suggest that the performance degradation observed in MMReD is not merely a scaling artifact but reflects a structural limitation of softmax attention in dense, integrative reasoning settings.

**Beyond single-needle retrieval.** Recent work has sought to move past simple NIAH formulations. Vodrahalli et al. (2024) introduced the Michelangelo benchmark to study the "short-circuiting" problem, where models exploit superficial correlations rather than genuinely using the entire context; its most demanding *latent list* setting requires tracking up to 20 relevant needles among distractors, yet state-of-the-art models still perform well above chance. In parallel, Bai et al. (2025b) proposed LongBench v2, which grounds evaluation in naturally occurring long contexts rather than synthetic haystacks, thereby improving ecological validity. In the video domain, Ben-Ami et al. (2025) make

the evidential demand of VideoQA benchmarks explicitly measurable via the *Minimum Required Frame-Set* (MRFS)—the smallest set of frames that must be fused to answer a question correctly. They show that prior VideoQA datasets have a mean MRFS of only 2.6–4.2, meaning most questions are resolvable from a handful of frames, whereas HERBench raises this to 5.5; nevertheless, even their hardest setting requires only a sparse subset of frames, and models still fail primarily due to retrieval and fusion deficits rather than an inability to reason over uniformly dense contexts. Nevertheless, Michelangelo, LongBench v2, and HERBench share the same structural property: the bulk of the context is still dominated by irrelevant or weakly informative content, and the task reduces to locating a sparse evidential subset. By contrast, MMReD is explicitly designed so that *all* context elements are densely informative, forcing models to integrate global patterns rather than locate sparse signals.

**Motivation for MMReD.** Prior benchmarks have advanced the study of long-context comprehension but converge on retrieval-centric formulations that do not directly test a model's ability to reason when all parts of the context are densely informative and must be integrated uniformly. MMReD is designed to fill this gap: it complements NIAH evaluations by systematically assessing reasoning in dense, non-retrieval-based multimodal contexts, and provides a diagnostic setting in which the theoretical limitations identified above can be measured empirically.

## 3    MMReD Benchmark

To address questions above, we create a visual environment with randomized and scalable state evolution and sufficient and diverse set of tasks to evaluate the key reasoning capabilities of models.

### 3.1    Design principles

We prioritize a minimalist visual representation. By avoiding visual complexity, we ensure that evaluation results reflect dense context multi-modal reasoning rather than visual-only perception, as in Wu et al. (2025). However, our design assumes that models possess basic OCR capabilities.

We also prioritize minimalist linguistic constructions. By doing so, we ensure that our evaluation remains focused on the same dense context multi-modal reasoning rather than language and instruction understanding, as in Kim & Schuster (2023); Vodrahalli et al. (2024). Despite linguistic simplicity, we ensure covering key reasoning categories with a diverse set of tasks.

Our benchmark is designed to scale in complexity on the context length axis. The dataset includes sequences of varying lengths, with longest sequences reaching 128 frames, where even the current best models start to systematically score $0\%$ accuracy in some tasks. With the increasing capabilities of models, the benchmark can be extended to larger contexts (e.g., 256 frames or beyond), ensuring that it remains a dynamic and evolving evaluation framework.

To ensure MMReD provides a fair and robust evaluation, we enforce strict controls on dataset generation. All generated sequences are unique, preventing unintended memorization or retrieval opportunities. The environment evolves randomly, preventing reliance on simple heuristics. And we balance generated answer distributions to prevent dominance of some frequency-biased methods.

### 3.2    Dataset

The MMReD dataset consists of structured sequences of frames representing an evolving environment. Each frame depicts a spatial arrangement of *characters* within predefined *rooms*. These sequences serve as context for various reasoning tasks, such as identifying patterns across time (e.g., "Which room was empty for fewer steps than the other rooms?"). See examples in Figure 1.

**Environment.** We define six distinct *rooms*: Kitchen, Bathroom, Garden, Office, Bedroom, and Hallway; and five *characters*: Sandra, Mary, Michael, John, and Daniel. At each time step, characters are assigned to rooms, with the possibility of multiple characters sharing a room or some rooms being empty. All characters remain present in the environment throughout the sequence.

**Sequence evolution.** Our dataset consists of multiple distinct sequences. A sequence consists of an initial state and a set of state changes. We generate sequences of varying lengths

$N \in \{1, 2, 4, 8, 16, 32, 64, 128\}$, with 1200 sequences per length. When $N = 1$, the sequence consists only of an initial state, with an empty set of state changes.

To generate a sequence of length $N$, we first sample an initial state by randomly (uniformly) assigning each character to one of the six rooms. At each of the following $N - 1$ steps, one randomly selected character moves to a different randomly selected room. We generate sequences iteratively and, if an obtained sequence already exists, repeat the process until a unique sequence appears.

**Scene construction.** Each state is visualized as a $512 \times 512$ pixel image, where rooms are depicted as rectangles located in a $2 \times 3$ grid, each named at its bottom. Characters are represented as colored circles, with their names displayed above. A character's presence in a room is shown by positioning its circle inside the corresponding rectangle. The state number is given at the bottom of the image.

**Tasks annotation.** Each sequence is paired with exactly one generated *question*, selected from a set of 24 question types. We generate 50 questions per type, resulting in 1200 question-sequence pairs per length $N$. To generate a question, we randomly sample rooms, characters, or a state number as needed for the question template. The *answer* is precomputed algorithmically using full access to the sequence and environment. If a correct singular answer does not exist, e.g., two characters satisfy the condition simultaneously, we repeat the sequence generation for this question.

## 3.3 TASKS

MMReD consists of two primary categories of questions: scene-referenced (NIAH) and dense context (DC) reasoning questions. Each category is designed to systematically evaluate different aspects of multi-modal reasoning.

**NIAH questions.** The first three sections of Table 1 correspond to the NIAH task. These questions are designed to be answered based on a *single image* within the sequence that satisfies the condition.

We created three subgroups of questions: First Appearance (FA), locating the earliest occurrence of a given entity or event; Final App (FI), focusing on the last occurrence; Frame X (FX), focusing on an explicitly specified frame. This division allows us to isolate and measure the lost-in-the-middle phenomenon (Wu et al., 2024), where models struggle to retrieve information from intermediate context positions.

Within each section, we construct multiple question templates that target core reasoning categories such as object tracking, counting, and spatial reasoning, similar to bAbI (Weston et al., 2016). We also diversify output types – rooms, characters, and integers – to test model robustness across different answer formats. This ensures that our evaluations remain representative of the model's overall reasoning ability.

Table 1: Overview of MMReD benchmark questions categorized by reasoning type. Each question template contains placeholders: [R], [C], and [X] represent randomly sampled rooms, characters, and step numbers, respectively. [comp] is a randomly chosen comparison phrase ("most" or "least amount of").

| ID | Question template |
|---|---|
| FA-FA-R | In which room did [C] first appear? |
| FA-CCFA-R | In which room was [C1] when [C2] first appeared in the [R]? |
| FA-FR-C | Who was the first to appear in the [R]? |
| FA-RCFA-C | Who was in the [R1] when [C] first appeared in the [R2]? |
| FA-NRFA-I | How many characters were in the [R1] when [C] first appeared in the [R2]? |
| FI-FA-R | In which room was [C] at the final step? |
| FI-CCFA-R | In which room was [C1] when [C2] made their final appearance in the [R]? |
| FI-LR-C | Who was the last to appear in the [R]? |
| FI-RCFA-C | Who was in the [R1] when [C] made their final appearance in the [R2]? |
| FI-NRFA-I | How many chars were in the [R1] when [C] made their final app in the [R2]? |
| FX-CF-R | In which room was [C] at step [X]? |
| FX-RF-C | Who was in the [R] at step [X]? |
| FX-CCF-C | Who was in the same room as [C] at step [X]? |
| FX-NCF-I | How many other characters were in the same room as [C] at step [X]? |
| FX-NE-I | How many rooms were empty at step [X]? |
| DC-RE-R | Which room was empty for the [comp] steps? |
| DC-WS-R | In which room did [C] spend the [comp] time? |
| DC-CR-R | Which room was crowded (three or more people) for the most steps? |
| DC-WHS-C | Who spent the [comp] time in the [R]? |
| DC-SA-C | Who spent the [comp] time alone in the rooms? |
| DC-ST-C | With whom did [C] spend the [comp] time together in the same room? |
| DC-SR-I | How many steps did [C] spend in the [R]? |
| DC-RV-I | How many different rooms did [C] visit? |
| DC-CC-I | How many times did a crowd (three or more people in one room) appear? |

**DC questions.** The final section of Table 1 introduces our novel **dense context (DC)** reasoning tasks. Unlike the NIAH questions, which require retrieval from a single image, these questions demand global and uniform attention to the entire sequence. We similarly diversify DC tasks to include the same reasoning categories and output types, resulting in nine question types.

**Problem formulation.** The final evaluation dataset consists of 9600 triplets (*sequence*, *question*, *answer*). The models' goal is to predict the *answer* given the input pair *sequence-question*. LVLMs receive sequences transformed to images, as described above, while LLMs receive a purely textual

representation. To assess model performance, we use *exact-match accuracy*, where a predicted answer is considered correct if and only if it matches the ground truth answer.

## 3.4 EVALUATION SETUP

**Models.** We evaluated approximately 30 multi-modal models on MMReD, including both open-source and proprietary models. Our primary focus was on image-specialized LVLMs, including Qwen2-VL (Wang et al., 2024a), Qwen2.5-VL (Bai et al., 2025a), InternVL-2.5 (Chen et al., 2024), InternVL-2.5-MPO (Wang et al., 2024c), MiniCPM-2.6-O (Yao et al., 2024), and LLaVA-Mini (Zhang et al., 2025b). Additionally, we evaluated video-oriented LVLMs, such as LLaVA-Video (Zhang et al., 2024b), Video-LLaMA3 (Zhang et al., 2025a), and Aria (Li et al., 2024). We also assessed a variety of LLMs: Qwen2 (Yang et al., 2024a), Qwen2.5 (Yang et al., 2024b), Qwen3 (Yang et al., 2025), and Qwen-Coder (Hui et al., 2024) families, distilled versions of the DeepSeek-R1 (Guo et al., 2025), and the QwQ (Team, 2025) reasoning model from the Qwen series. Among the proprietary models, OpenAI's GPT-4o (OpenAI, 2024a) ('2024-11-20') and GPT-4o-mini (OpenAI, 2024b) ('2024-07-18') were tested using their official APIs.

**Benchmark representations.** LLMs require a textual representation of our benchmark. Thus, we transform sequences into JSON files, explicitly writing the frame numbers and corresponding characters locations. Furthermore, video-oriented LVLMs use different methods for sampling frames from the input visual sequence. So we fed MMReD images to video-oriented LVLMs as a result of frame sampling, ensuring an identical amount of input information to all models.

**Fine-tuning.** We also tested whether fine-tuning can help generalize to unseen context lengths. Both SFT and GRPO (Shao et al., 2024) training were performed on the same dataset spanning $N = [1, 2, 4, 8, 16]$ sequence lengths, 200 samples per each task of the benchmark per sequence length. We used Qwen2.5-7B-Instruct (Yang et al., 2024b) and Falcon3-Mamba-7B-Instruct (Team, 2024) models for SFT and DeepSeek-R1-Distill-Qwen-7B (DeepSeek-AI et al., 2025) for GRPO.

Technical details, such as system prompt, output formats, generation parameters (consistent across all models), model hosting, fine-tuning hyperparameters, and resources, are provided in Appendix.

## 4 RESULTS AND ANALYSIS

### 4.1 MAIN RESULTS

Average performance across all tasks of models, including both multi-modal and text-only variants, is presented in Figure 3a. Results for each model and model group are given in Supplementary materials. Based on these results, we draw several key conclusions.

**MMReD is a universal and cross-modal task for dense reasoning.** The performance of all evaluated models begins to degrade significantly for sequence lengths exceeding 32 steps. Which is even more evident for the DC tasks (Figure 3b). This degradation rate strongly corresponds to the number of parameters – larger models demonstrate a greater ability to maintain accuracy in constructing logical chains over longer sequences. Furthermore, reasoning-specialized LLMs not only exhibit superior initial performance but also show greater robustness to increases in the length of the reasoning chain. It is worth noting that the Qwen2.5-Coder model, fine-tuned for coding-specific tasks, underperformed compared to the original Qwen2.5. This suggests that training on coding tasks alone does not sufficiently reinforce the ability to construct and infer logical reasoning chains.

**Multimodal instruction tuning impairs long-context understanding.** Additionally, LVLMs struggle to utilize visual context effectively, even when the context length remains within the claimed supported limits. For instance, InternVL2.5, which reports a context length of 16,384 tokens and uses 256 tokens to encode a single image, should handle tasks involving up to 64 images with optimal accuracy. But we observe a decline in performance starting from 16 images. For sequences exceeding 64 images, the models' ability to generate coherent reasoning chains deteriorates to the point where extracting a final answer becomes unfeasible.

**Impact of LLM reasoning on long-context retention.** Proprietary GPT-4o and its 'mini' version outperform most open-source models in both textual and multi-modal representations of the benchmark. Their accuracy on short sequences is nearly perfect, and as the length of the sequence

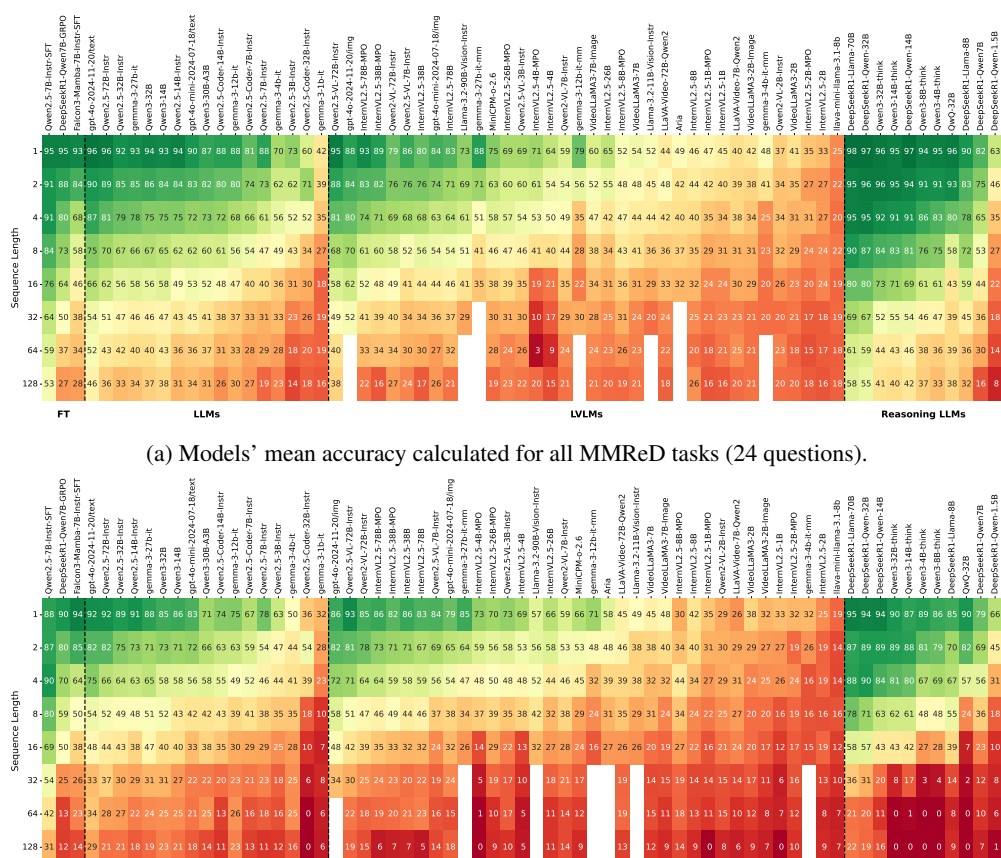

(a) Models' mean accuracy calculated for all MMReD tasks (24 questions).

(b) Models' mean accuracy calculated only for dense context (DC) type of tasks.

Figure 3: Models' mean accuracy on MMReD across all sequence lengths, grouped by model type: fine-tuned LLMs, general-purpose LLMs, LVLMs, and reasoning LLMs. Cell color indicates performance (green – high, red – low), with models ranked by average score within each group.

increases, the degradation in quality is smoother. However, DeepSeek R1, distilled to LLaMA-70B and Qwen2.5-32B outperform GPT-4o by 2-4% initially and considerably more at longer sequence lengths, as shown in Supplementary materials. This highlights the substantial contribution of LLM reasoning capabilities to retaining long preceding contexts and deriving correct results.

## 4.2 TASK ABLATIONS

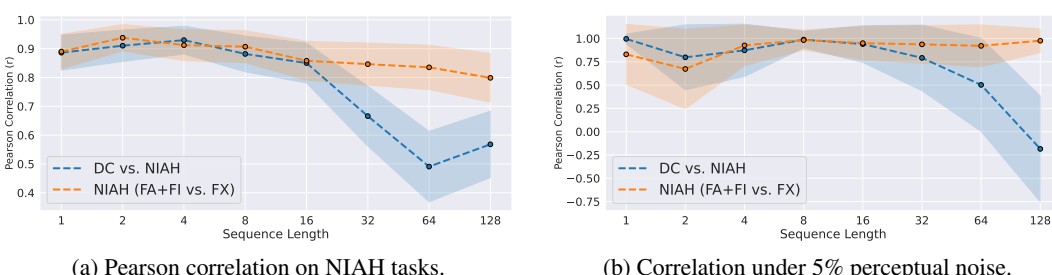

(a) Pearson correlation on NIAH tasks.

(b) Correlation under 5% perceptual noise.

Figure 4: Analysis of model performance: (a) Self-consistency vs. DC tasks and (b) Performance under noise conditions.

**Isolating DC reasoning.** Figure 4a illustrates the breakdown in correlation between performance on retrieval-based (NIAH) and DC reasoning tasks as context length increases. We calculate Pearson correlation of model scores averaged for every task within selected task groups. In the first experiment, we select DC and NIAH groups; in the second experiment, we select FA plus FI and FX groups. While Pearson correlation between different NIAH subsets remains consistently high (around 0.9), the correlation between NIAH and DC performances notably declines to approximately 0.5–0.7 at 32 frames and beyond.

Our findings closely align with those reported in BABILong (Kuratov et al., 2024): correlation between the models' ranking on their long-context benchmark and short-context one drops from 0.9 to 0.6 when context length increases. These findings suggest that models successful in NIAH-style retrieval do not necessarily generalize to the proposed DC reasoning scenarios. And thus MMReD allows assessing a conceptually different capability, which we call **dense context** reasoning.

**Perceptual complexity.** Perceptual ambiguity, e.g., occlusion, visual noise, recognition errors, plays a major role in real-world multimodal reasoning. However, MMReD is specifically designed to isolate reasoning capabilities by minimizing perceptual complexity.

To test whether this abstraction limits generalizability, we conducted an additional ablation where we introduced controlled perception noise into MMReD. Specifically, we modified the environment state by adding synthetic occlusion-like errors (e.g., randomly relocating an entity to an incorrect room in one frame) at a 5% rate. We then evaluated LLM performance on both NIAH and DC questions; see Figure 4b.

The results show a uniform drop in performance proportional to the error rate, but crucially, the relative gap between LC and NIAH questions, the key trend in Figure 4a, remain intact. While we conducted this test on LLMs, the insight generalizes to LVLMs as well. In our setting, LVLMs and LLMs show a consistent delta, likely due to (i) the token budget shift toward visual inputs, leaving fewer tokens for reasoning, and (ii) catastrophic forgetting from vision-domain fine-tuning. Thus, solving dense context reasoning in LLMs transfers naturally to LVLMs.

**Symbolic environment.** We evaluated MMReD under an abstract symbolic projection (5 locations coded as L1–L5, 6 entities coded as E1–E6) to test whether results depend on the visual and structural variations. Model rankings and overall performance trends are preserved under the symbolic representation: Pearson correlations between standard and symbolic environments remain high across sequence lengths (see Appendix Table 4), indicating that observed DC vs NIAH gaps are not surface-specific to the visual "rooms and characters" instantiation. We therefore interpret MMReD as probing a structural DC reasoning challenge rather than an artifact of the chosen visual motif.

In summary, we show that adding controlled perceptual noise nor structural variations do not alter the key conclusions of MMReD. **Our benchmark allows studying dense reasoning in isolation**, and can be extended to real-world scenarios by progressively incorporating perceptual challenges.

## 4.3 MODEL ABLATIONS

The core motivation for our experiments stemmed from the question: *What components or approaches enable LVLMs to understand long visual sequences more effectively?* To address this, we formulated a series of research questions focusing on architectural components, dataset characteristics, and training procedures. These questions are systematically explored in this section.

**Text-only (LLM) vs Multimodal (LVLM) performance gap.** We computed the relative performance gap between text-only and multimodal variants as $\frac{\text{LLM}-\text{LVLM}}{\text{LVLM}} \times 100\%$. The gap varies with model scale and context length: text representations generally outperform multimodal ones at mid context lengths, while smaller models sometimes invert at extreme N (Table 2). This pattern suggests that text encodings reduce perceptual token-budget pressure, while for smaller models compressed visual encodings can be advantageous.

**Fine-tuning is not enough for generalization.** Both fine-tuned Transformer and Mamba architectures demonstrate the same decline in performance as benchmarked zero-shot models. GRPO, being a promising way to bootstrap reasoning, is performing even worse than fine-tuned Transformer. Overall, fine-tuning generalizes a bit better than best regular LLM and worse than reasoning LLMs, in particular, on tasks involving heavier arithmetic.

Table 2: Relative performance gap (%) between text-only and multimodal models across sequence lengths. Positive values indicate a text-only advantage.

| Model | $N=1$ | $N=2$ | $N=4$ | $N=8$ | $N=16$ | $N=32$ | $N=64$ | $N=128$ |
|---|---|---|---|---|---|---|---|---|
| Qwen2.5-3B-Instruct | 6.7 | 4.1 | -2.3 | -5.7 | -12.0 | -23.2 | -32.2 | -37.5 |
| Qwen2.5-7B-Instruct | 2.3 | -4.0 | -11.3 | -9.1 | -1.8 | -9.6 | -3.9 | -19.1 |
| Qwen2.5-72B-Instruct | 0.6 | 1.2 | 0.6 | 2.9 | 5.7 | 3.9 | 6.2 | -3.8 |
| GPT-4o | 9.8 | 7.1 | 8.3 | 6.0 | 7.2 | 3.8 | – | – |
| GPT-4o-mini | 7.2 | 12.6 | 14.8 | 14.7 | 21.0 | 27.6 | 32.5 | 27.6 |
| Mean | 11.0 | 16.8 | 35.1 | 35.3 | 34.1 | 12.5 | 0.7 | -8.2 |

Additionally, we performed a controlled In-Context Learning (ICL) ablation by appending five solved examples (from $N \leq 16$) to each prompt. Across LLMs (4B–32B) we observe a mean relative improvement of 13.8%; mid-range models gain up to 23% (Table 3). Although ICL yields consistent gains, it is substantially smaller than SFT on the same mid-range models (57.7%) and GRPO (27.8%), and none of these approaches fully resolve dense-context failures at large $N$.

Table 3: Relative mean improvement (%) over baseline across sequence lengths for ICL, GRPO, and SFT. The last column reports the relative improvement on the full dataset.

| Method | N=1 | N=2 | N=4 | N=8 | N=16 | N=32 | N=64 | N=128 | Full dataset |
|---|---|---|---|---|---|---|---|---|---|
| ICL | 13.4 | 16.9 | 17.9 | 17.2 | 13.8 | 7.5 | 7.7 | 12.5 | 13.8 |
| GRPO | 14.6 | 16.9 | 23.2 | 36.6 | 44.9 | 37.1 | 26.5 | 72.0 | 27.8 |
| SFT | 8.0 | 24.2 | 49.1 | 76.9 | 89.8 | 109.2 | 99.8 | 178.5 | 57.7 |

**Model Parameter Count.** To investigate the effect of model size on MMReD response quality, we compared the evaluation results of both LVLMs and LLMs, focusing on three representative model families (Figure 5). The results from the Qwen2.5-VL and InternVL2.5 series reveal a clear trend: accuracy in question answering improves as the number of parameters increases. However, for the InternVL2.5-38B and 78B models, the performance gap narrows significantly, becoming much smaller compared to the differences observed among smaller models. A similar pattern is evident in the results of DeepSeek-R1 distilled to LLaMA-70B and Qwen-32B. These findings suggest that, for both visual and text-only models, the number of parameters strongly influences the ability to process long contexts. However, this dependence diminishes as model size increases, with visual reasoning quality eventually plateauing beyond a certain threshold.

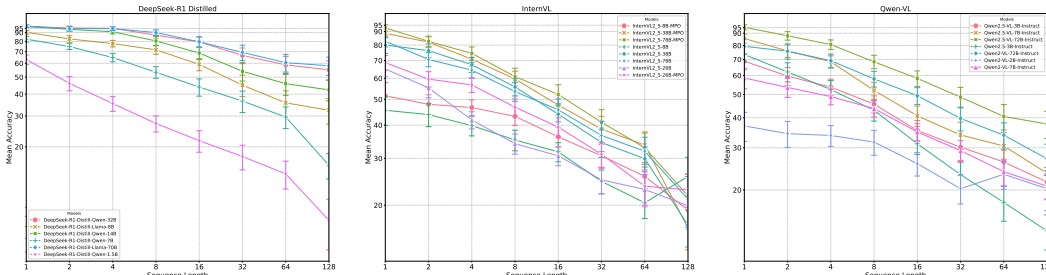

Figure 5: Impact of the number of model parameters on the ability to successfully infer a response using a long visual or text context from MMReD.

**Multimodal Adapter Type.** We evaluated the efficiency of LVLMs using different adapter types: multi-linear layer adapter (Qwen2.5-VL-7B-Instruct (Bai et al., 2025a)), QFormer (MiniCPM-o-2.6 (Yao et al., 2024)), and cross-attention (Llama-3.2-11B-Vision-Instruct (Chu et al., 2024)). The results in Figure 11 show that LLaMA does not provide high performance. The results of Qwen2.5-VL and MiniCPM are roughly the same: both models perform poorly on DC-CC-I and DC-SR-I questions, but MiniCPM performs better on FI-FA-R ones. Moreover, Qwen2.5 has low scores for short sequences in certain types of questions.

**Frame Pooling Methods in Video-Specific Models.** We also compare performance of the different video-specific models: InternVL-2.5 (Wang et al., 2024c), Aria (Li et al., 2024), VideoLLaMA-3 (Zhang et al., 2025a), and LLaVA-Video-7B-Qwen2 (Zhang et al., 2024b). As shown in Figure 12, VideoLLaMA-3 provides the best performance. Aria performs better than InternVL-2.5. Moreover, InternVL-2.5 has a considerable drawdown for all sequence lengths in certain types of questions.

**Training Data Volume and Composition.** We compare two types of VideoLLaMA3 models: VideoLLaMA3-Image, fine-tuned on both image-text and video data, and VideoLLaMA3, which undergoes video-centric fine-tuning. Both variants use alignment (media paired with captions) and instruction (question answering) data. As can be seen in Figure 13, the VideoLLaMA3-Image family performs better on straightforward questions with short sequences. However, its performance declines on longer sequences, particularly for question types such as DC-SR-I and DC-SA-C.

We argue that incorporating video pretraining without textual instructional data on the last training stage negatively impacts the model's performance on our benchmark. As shown in the Figure 3a, models from the LLaVA-Video family, even 72B ones, perform worse than image-focused or general purpose LVLMs.

Possible directions of solving the challenge of dense context reasoning include, but are not limited to, examining the potential of architectural innovations specifically designed for long-context processing, overcoming problems of the attention sinks and similar known expressivity issues of RNNs, and incorporating uncertainty quantification (Fadeeva et al., 2024; 2023; Vashurin et al., 2025) to make reasoning more steerable and not materialize false information in the decoded logits; building on existing approaches of test-time compute scaling alongside rigorous RL goal-based pre-training.

## 5 CONCLUSION

In this paper, we introduced MMReD, a novel benchmark designed to systematically assess long-context reasoning capabilities of LLMs and LVLMs. By focusing on both scene-referenced (NIAH) and dense context tasks, MMReD provides a comprehensive evaluation framework that challenges models to process extended multimodal sequences and construct complex reasoning chains.

Our results demonstrate that current LLMs and LVLMs struggle significantly with long-context understanding, with performance degrading rapidly as context length increases beyond 32 images. Notably, reasoning-specialized LLMs exhibit superior retention of long contexts compared to standard models, highlighting the importance of targeted fine-tuning and architectural innovations in enhancing long-context comprehension.

Model size positively correlates with long-context performance, but the benefits diminish at larger scales, suggesting architectural limitations rather than parameter count as a key bottleneck. Additionally, our analysis of multimodal adapter types and frame pooling strategies revealed that the architectural components choice critically affects model's ability to reason over long visual sequences.

Unexpectedly, video-oriented LVLMs and multimodal instruction tuning are not beneficial for long-context reasoning. The observed performance decline for these models on extended sequences suggests that current video pretraining does not sufficiently address the challenges of long-context tasks.

Our ablation studies confirmed that MMReD's conclusions are robust to potential confounding factors such as the lost-in-the-middle phenomenon. By maintaining unique and randomly evolving environments, the benchmark ensures that performance degradation reflects genuine limitations in long-context reasoning rather than artifacts of data structure or task formulation.

In conclusion, our findings underscore the need for more effective architectural modifications, training paradigms, and evaluation benchmarks tailored specifically for long-context reasoning. MMReD serves as a critical step forward in this direction, providing a rigorous, scalable, and unbiased framework for assessing and advancing long-context understanding in both LLMs and LVLMs. Future work may explore integrating hierarchical memory mechanisms, improved multimodal fusion techniques, and more diverse pretraining data to further enhance long-context reasoning capabilities.

## REPRODUCIBILITY STATEMENT

We release our training and evaluation codebase[1] and dataset [2] in the anonymous repositories.

## USE OF LARGE LANGUAGE MODELS.

In preparing this manuscript, we made use of large language models (LLMs) to aid in polishing the writing. Specifically, we used an LLM to (i) suggest alternative phrasings for certain sections (Related Work and Conclusion), (ii) merge fragmented paragraphs into a more coherent narrative, and (iii) check stylistic clarity and consistency across sections. All scientific contributions, including the design of the benchmark, experimental setup, implementation, analysis, and conclusions, were developed entirely by the authors. The LLM did not generate new content beyond language-level editing and restructuring, and its usage did not rise to the level of a contributing author.

## ACKNOWLEDGMENTS

Innopolis University authors were supported by the Research Center of the Artificial Intelligence Institute at Innopolis University. Financial support was provided by the Ministry of Economic Development of the Russian Federation (No. 25-139-66879-1-0003).

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

## A  DETAILED SYMBOLIC ENVIRONMENT VALIDATION

To test whether MMReD results depend on the specific visual "rooms & characters" projection, we introduced an abstract symbolic variant (5 locations $\times$ 6 entities). We computed the Pearson correlation coefficient $r$ between model performance in the standard environment and the symbolic environment across sequence lengths.

Table 4: Correlation of model rankings between the original projection and a novel symbolic projection (5 locations $\times$ 6 entities). High correlations indicate ranking stability across projections, supporting generality of observed trends.

| Seq len $N$ | 1 | 2 | 4 | 8 | 16 | 32 | 64 | 128 |
|---|---|---|---|---|---|---|---|---|
| Pearson $r$ | 0.98±0.09 | 0.95±0.16 | 0.80±0.30 | 0.72±0.35 | 0.69±0.36 | 0.89±0.23 | 0.86±0.25 | 0.83±0.28 |

We observe consistently high correlation ($r > 0.7$ for most context lengths), peaking at $r > 0.98$ for short contexts. This indicates that model rankings remain largely stable across projections. The results suggest that MMReD captures a structural dense-context reasoning challenge rather than overfitting to surface-level visual semantics.

We further analyze the relative performance shift when moving from the standard to the symbolic environment. We compute the relative change: $\Delta = \frac{\text{Symbolic} - \text{Original}}{\text{Original}} \times 100\%$.

Table 5: Relative performance change ($\Delta$, in %) when shifting from the standard to the symbolic environment across sequence lengths. Positive values indicate improvement in the symbolic domain.

| Model | $N=1$ | $N=2$ | $N=4$ | $N=8$ | $N=16$ | $N=32$ | $N=64$ | $N=128$ |
|---|---|---|---|---|---|---|---|---|
| Qwen3-30B-A3B | −16.0 | −6.3 | −1.3 | 0.9 | −6.7 | −5.7 | −5.0 | −3.5 |
| Qwen3-Next-80B-A3B | −18.8 | −4.0 | 1.4 | 4.2 | −4.8 | −7.0 | 3.0 | 2.7 |
| Qwen3-4B | −12.3 | 5.9 | 8.1 | 9.7 | 8.7 | −2.6 | −5.1 | 7.1 |
| Qwen3-8B | −6.9 | −5.9 | −10.1 | −6.8 | −21.3 | −10.5 | −7.9 | 9.6 |
| Qwen3-14B | −15.5 | −3.2 | 3.8 | 10.3 | 12.0 | 8.9 | 6.3 | 18.3 |
| Qwen3-32B | −16.2 | −11.6 | −14.0 | −13.0 | −17.1 | −9.3 | −13.4 | −4.6 |
| mean ± std | −14.3 ± 4.2 | −4.2 ± 5.9 | −2.0 ± 8.3 | 0.9 ± 9.2 | −4.9 ± 13.4 | −4.4 ± 7.1 | −3.7 ± 7.4 | 4.9 ± 8.6 |

The results reveal a distinct dichotomy between initial grounding and long-context tracking:

**Semantic Grounding Effect.** At sequence length $N = 1$, we observe a substantial performance drop in the symbolic environment ($-14.3$). This suggests that natural language semantics (e.g., labels such as "Kitchen") facilitate initial grounding compared to abstract tokens. The performance decrease indicates that early-stage reasoning benefits from semantic priors embedded in natural language representations.

**Convergence at Dense Contexts.** As sequence length increases, the performance gap narrows considerably. At extreme context lengths ($N = 128$), the mean gap becomes positive ($+4.9$), indicating that symbolic representations may even improve performance in dense-context regimes. This suggests that long-context difficulty primarily stems from structural tracking complexity rather than semantic interpretation. In certain models (e.g., Qwen3-14B), abstract symbols may reduce distraction from semantic priors, allowing the model to focus more directly on state transitions.

## B  LOST-IN-THE-MIDDLE PHENOMENON

This phenomenon refers to the tendency of transformer-based models to struggle with retrieving information from the middle of long contexts. Since our benchmark involves reasoning over extended sequences, we evaluated whether such biases affect our results.

To investigate this, we focus on frame-referenced questions (FX, see Table 1), where a model must extract information from a specific step in the sequence. This includes five question types, where we can calculate the referred image depth by extracting the "step X" number and dividing it by the sequence length.

By averaging performance across all LVLMs and plotting accuracy as a function of the frame position (Figure 7), we observe no clear pattern indicative of lost-in-the-middle degradation. If the latter phenomenon were significantly impacting our benchmark, we would expect accuracy to consistently drop for middle-positioned frames compared to the beginning and end, which is not the case. When analyzing individual model performances, we observed similar trends – no systematic drop in accuracy for middle-positioned frames.

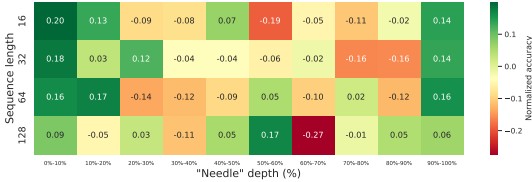

Figure 7: Analysis of the lost-in-the-middle effect on MMReD frame-referenced questions. The x-axis represents the relative position of the target frame within sequence; the y-axis shows the normalized accuracy difference from the mean.

This suggests that our benchmark and its conclusions remain independent of lost-in-the-middle biases, at least within the tested context lengths. And we believe that longer context lengths may be required before this effect becomes a major factor in our setting. The results suggest that longer context lengths may be required before the LITM effect becomes a major factor in our setting.

## C  MODELS' PERFORMANCE

Below, we present detailed heatmaps of model performance across all MMReD task types and sequence lengths. Figure 8 focuses on top-performing reasoning models. A clear trend emerges: dense reasoning (DC) tasks are consistently more challenging than their NIAH-based counterparts, this gap is particularly evident in the QwQ model. For proprietary models (Figure 9), the contrast is less visually evident but persists when examining average performance across tasks.

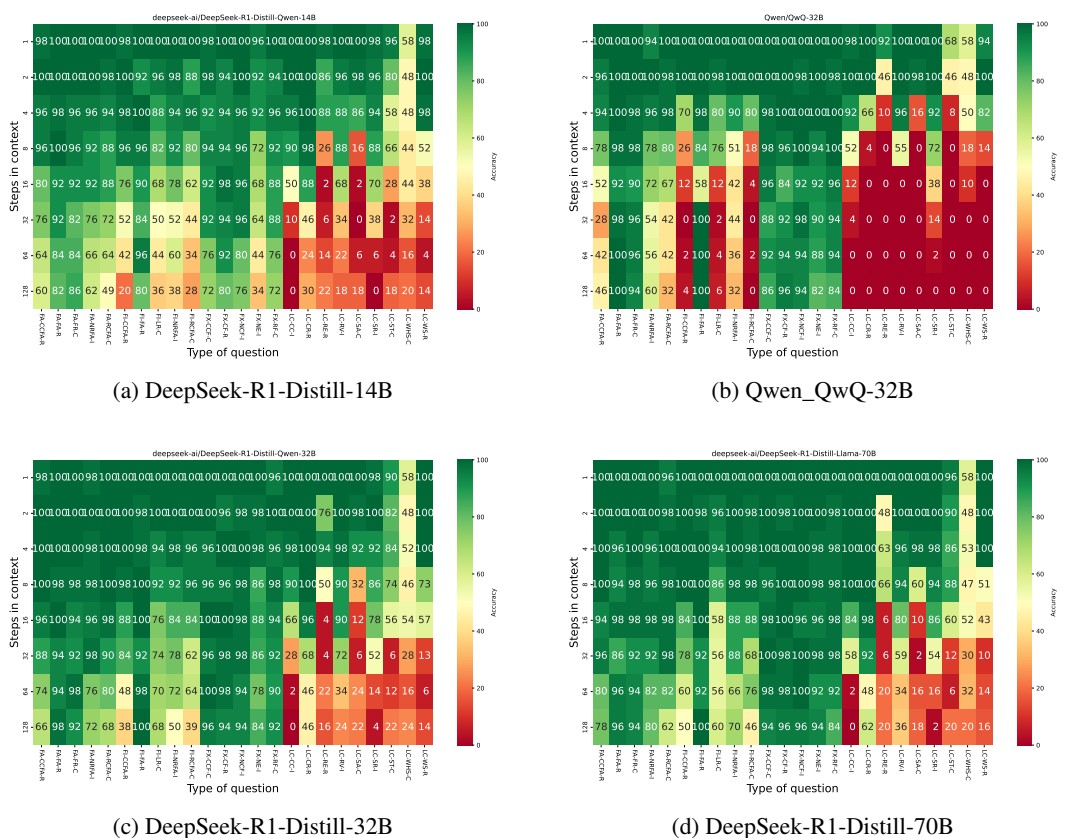

(a) DeepSeek-R1-Distill-14B

(b) Qwen_QwQ-32B

(c) DeepSeek-R1-Distill-32B

(d) DeepSeek-R1-Distill-70B

Figure 8: Performance of the best reasoning models.

Secondly, we found that including a diverse set of question types is essential, as model performance varies across conceptually distinct tasks. For example, despite the near-identical formulation of first appearance (FA) and last appearance (FI) questions, models consistently perform worse on average on FI tasks (e.g., Figure 8d). This suggests that models struggle to distinguish between similar temporal trajectories in task formulation, especially when we see questions referring to a concrete step (FX), which are phrased in a similar way, being solved with notably higher accuracy. In addition, our benchmark includes tasks with gradually increasing difficulty, with DC-WHS-C (Who spent the most time in the [R]?) and DC-RE-R (Which room was empty for the most steps?) among the most challenging. The latter highlights a recurring weakness in count-based reasoning.

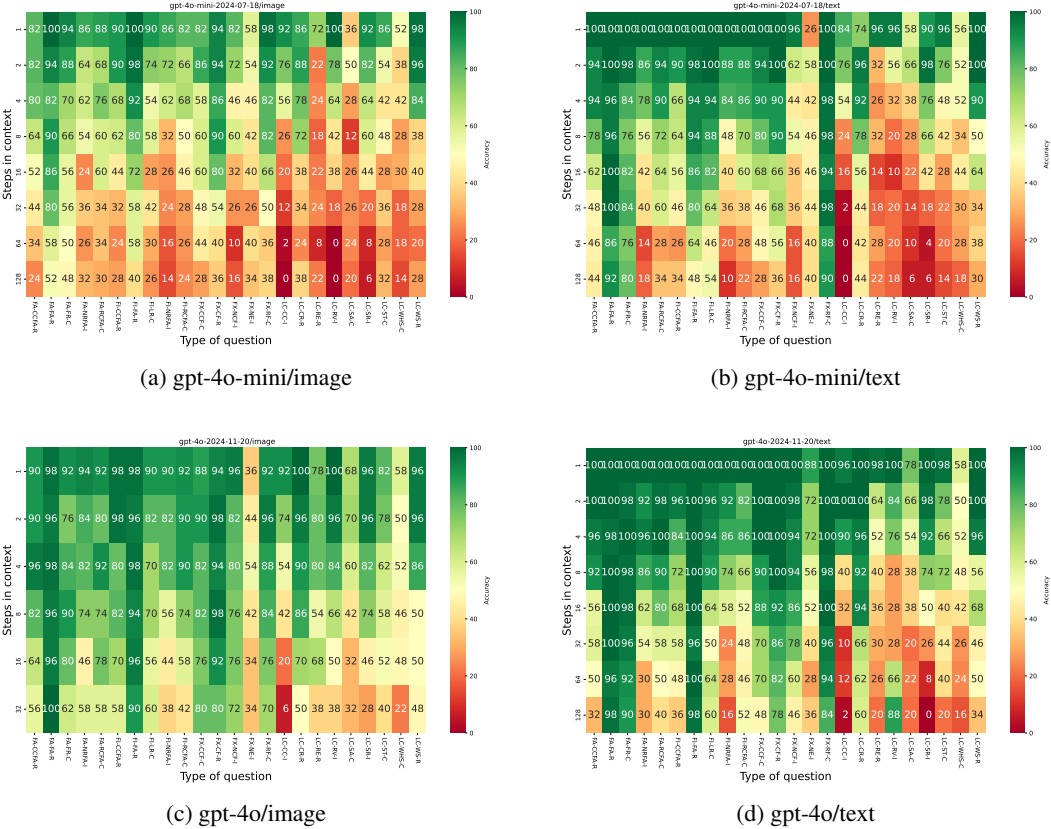

(a) gpt-4o-mini/image        (b) gpt-4o-mini/text

(c) gpt-4o/image        (d) gpt-4o/text

Figure 9: Performance of the OpenAI proprietary models.

As discussed in Section 4.3, various fine-tuning strategies fail to improve performance on our benchmark with their results detailed in Figure 10. Supplementing Section 4.3 further, studies on different pooling methods are provided in Figure 12. Finally, we report results for video-specific architectures in Figures 11 and 13.

## D  EVALUATION DETAILS

**Inference format and parameters** The MMReD questions were formulated as open-ended question-answer tasks. We provided a generic problem description and answer format in the system prompt (Table 2).

To standardize output generation, we applied a structured generation approach to produce consistent answer structures in a JSON format, {"answer": <answer_type>}, where <answer_type> was an enum or integer number depending on the expected answer type (C, R, I in Table 1). We utilized outlines library (Willard & Louf, 2023) for structured generation, which involves converting the JSON response structure into a regular expression, which is further used to construct a finite state machine to guide LLM generation by adding bias to logits. We used

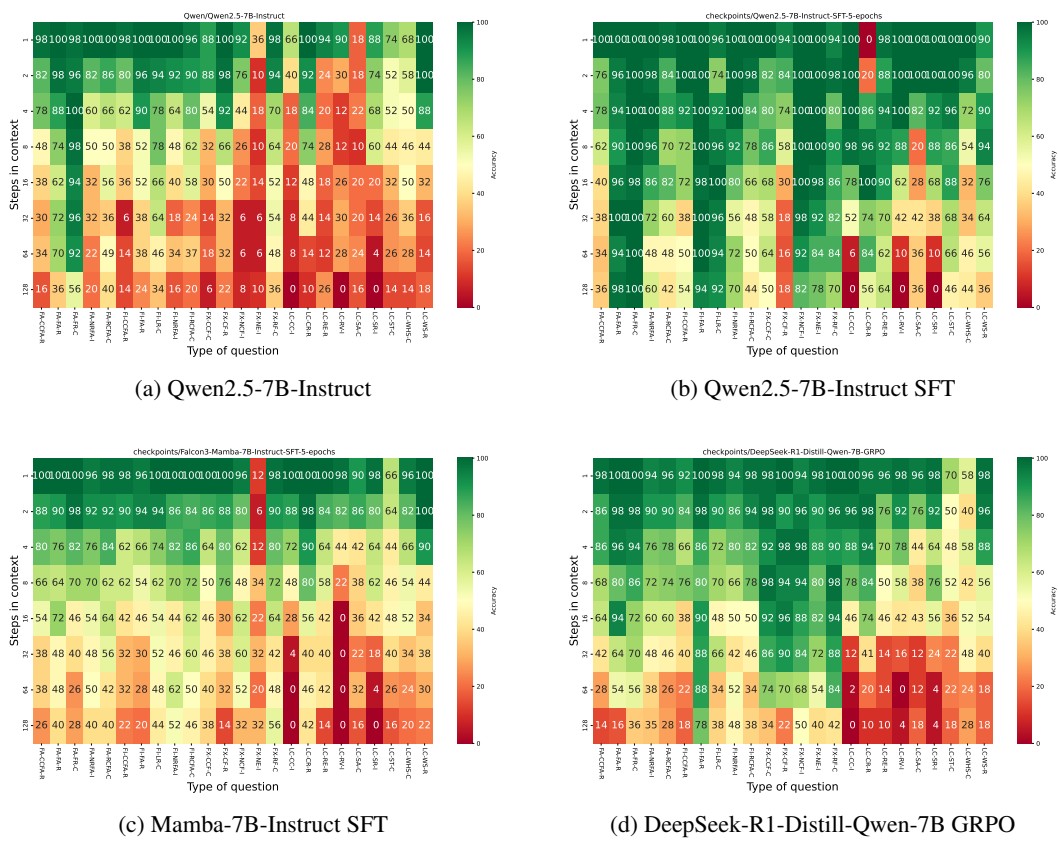

(a) Qwen2.5-7B-Instruct

(b) Qwen2.5-7B-Instruct SFT

(c) Mamba-7B-Instruct SFT

(d) DeepSeek-R1-Distill-Qwen-7B GRPO

Figure 10: Performance on the fine-tuned models and base model before fine-tuning.

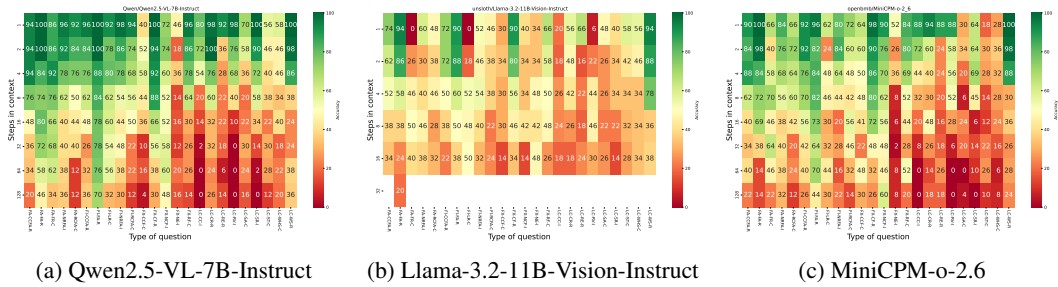

(a) Qwen2.5-VL-7B-Instruct

(b) Llama-3.2-11B-Vision-Instruct

(c) MiniCPM-o-2.6

Figure 11: Performance of the models with different types of image pooling.

the `lmdeploy` (Contributors, 2023) and `VLLM` (Kwon et al., 2023) packages for efficient serving of models.

We preserve the default configuration parameters of all selected LVLMs, except we extend the maximum sequence length and increase the positional embeddings to avoid truncating the input sequence. The generation parameters were also consistent across all models: the generation temperature was set to 0, the number of beams was set to 1, and the number of tokens was limited to 50. For LLMs with reasoning, we increase the generation temperature to 0.7 and token limit to 2048.

We used at most $4 \times$ A100 80GB in our local evaluations, and if model exceeded a time limit of 600 seconds or returned out-of-memory error, we dropped evaluation of the current and larger lengths (e.g., in cases of Aria and Llama-3.2-Vision).

**Fine-Tuning** We also ablated whether fine-tuning can help generalize to unseen context lengths. Both SFT and GRPO (Shao et al., 2024) training were performed on the same dataset spanning

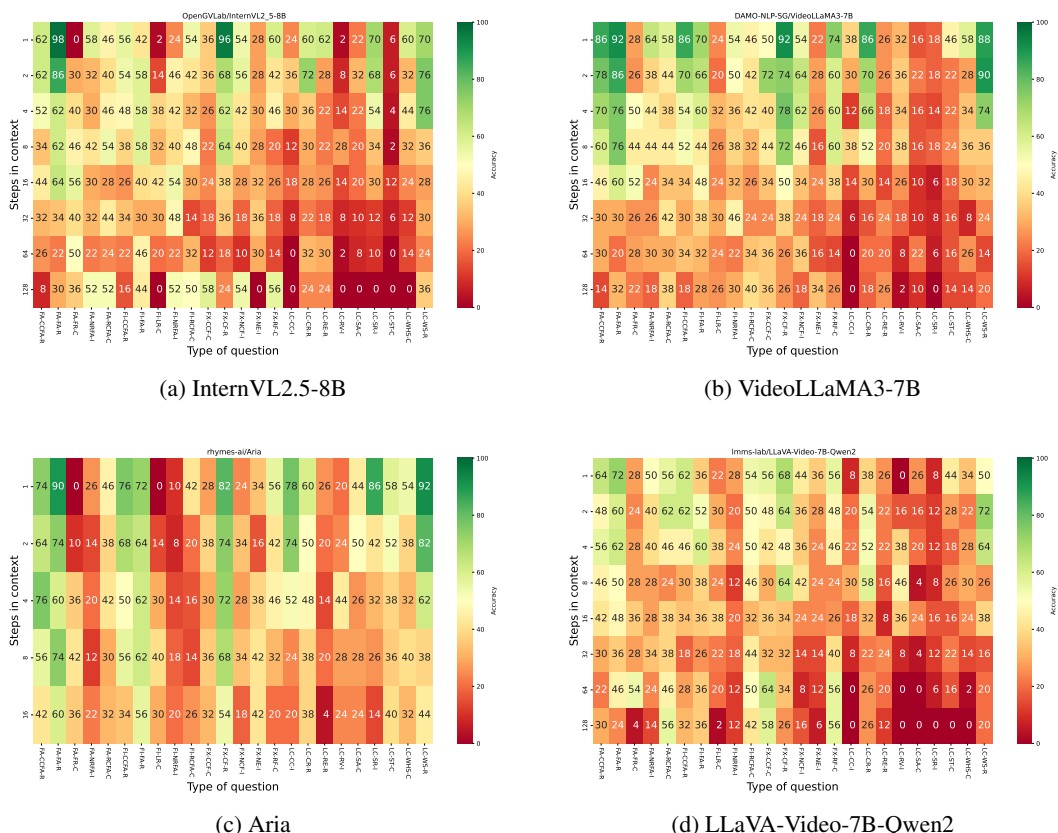

(a) InternVL2.5-8B

(b) VideoLLaMA3-7B

(c) Aria

(d) LLaVA-Video-7B-Qwen2

Figure 12: Performance of the different types of video models.

$N = [1, 2, 4, 8, 16]$ sequence lengths, 200 samples per each task of the benchmark per sequence length. We used Qwen2.5-7B-Instruct (Yang et al., 2024b) and Falcon3-Mamba-7B-Instruct (Team, 2024) models for SFT and DeepSeek-R1-Distill-Qwen-7B (DeepSeek-AI et al., 2025) for GRPO to help bootstrap reasoning abilities. In SFT, we used a global batch size of 128, learning rate of $3 \times 10^{-4}$, and five training epochs; GRPO was trained with $\varepsilon = 0.2$, $\beta = 0.1$, 12 generations per sample, global batch size of 168, and one epoch of training. We used linear warmup of 50 steps with a constant schedule after warmup and ScheduleFree AdamW (Defazio et al., 2024) optimizer. The gradient norm was clipped to $0.25$.

For GRPO rewards, we used a combination of answer correctness and format rewards, enforcing the model to follow an XML template
`<think>{thinking}</think> <answer>{"answer": <answer_type>}</answer>`
and rewarding for exact matches with ground truth answers.

We applied parameter-efficient fine-tuning via LoRA (Hu et al., 2021) with a rank of 32, alpha of 32, and dropout of $0.05$ in both scenarios. Flash Attention (Dao et al., 2022; Dao, 2024) was enabled for attention models, and mixed precision training was conducted with bfloat16. Mamba training was performed with gradients accumulating in float32 and using efficient CUDA kernels for selective scan operation from the Mamba codebase (Gu & Dao, 2024; Dao & Gu, 2024).

As an evaluation metric, we utilized the accuracy of the exact match between the generated response and the target value. Model predictions that did not follow the specified response format or from which the response field could not be extracted were marked as incorrect.

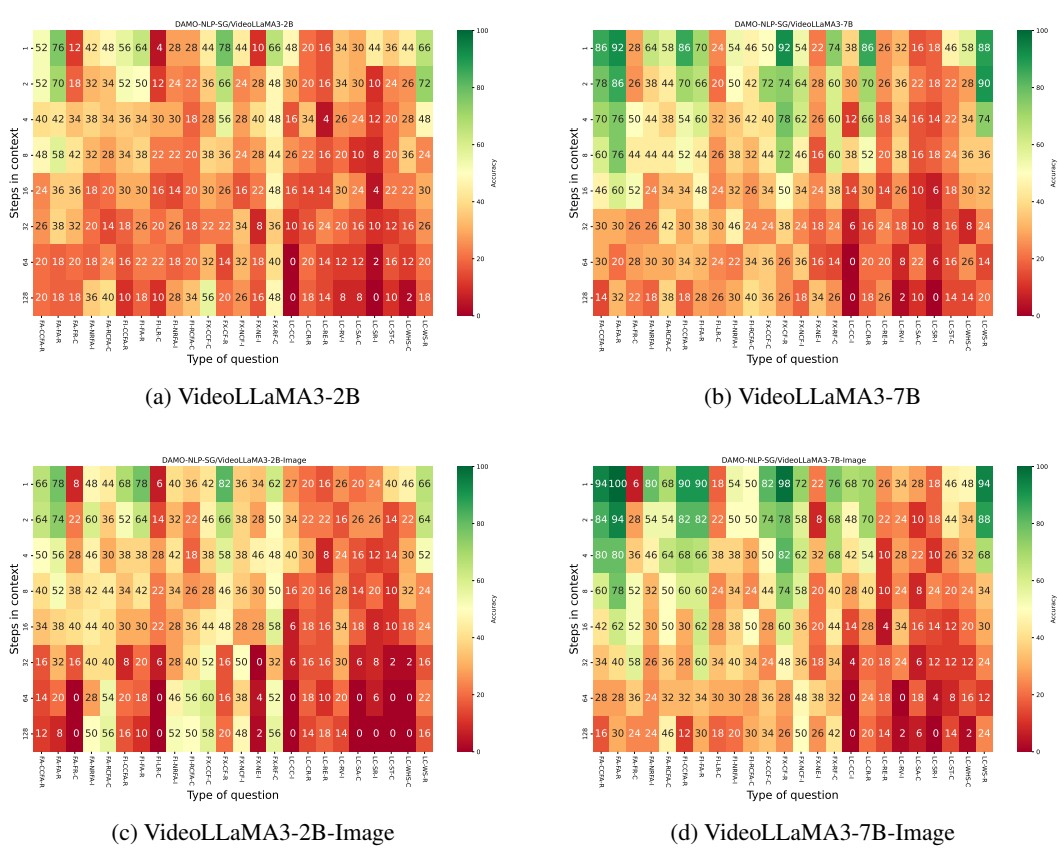

(a) VideoLLaMA3-2B

(b) VideoLLaMA3-7B

(c) VideoLLaMA3-2B-Image

(d) VideoLLaMA3-7B-Image

Figure 13: Results across different models in the VideoLLaMA-3 family.

---

**System Prompt**

---

```
You are an assistant that analyzes sequences of human agents
moving in an environment.  Format your response as a following
json:
{ "answer":  <value> }, where <value> is:
- A **single room name** (e.g., 'Kitchen') for location
answers.
- A **number** (e.g., '3') for counting answers.
- A **single person name** (e.g., 'Michael') for people
answers or 'Nobody' if no person satisfies given conditions.
```

---

**System Prompt for reasoning-based models**

---

```
You are a helpful AI Assistant, designed to provided
well-reasoned and detailed responses.  First think about the
reasoning process and then provide the user with the answer.
Format your final answer with a {"answer":  <value>}, where
<value> is:
- A **single room name** (e.g., 'Kitchen') for location
answers.
- A **number** (e.g., '3') for counting answers.
- A **single person name** (e.g., 'Michael') for people
answers or 'Nobody' if no person satisfies given conditions.
```

---

Table 2: System prompts used during inference.

## E  REASONING EXAMPLES

In general, we find that reasoning improves performance: when given sufficient token space, it often guides the model to the correct answer by helping it locate the relevant step and correct errors that non-reasoning models typically make (see example in Table 3).

However, in dense context (DC) tasks, starting at around 32-frame sequences, reasoning models begin to shift their output toward a step-by-step, JSON-like listing of observations. This introduces two key issues. First, a model may make a mistake when processing one step, and then proceed to build on that false assumption, effectively compounding and grounding on its error, as in example in Table 4. Second, enumerating all steps consumes a large portion of the token budget, increasing the risk that the model fails to produce a final answer within the predefined 768-token generation limit (a threshold, commonly sufficient for solving math problems (Cobbe et al., 2021), and thus adequate for our simpler reasoning tasks).

---

**DeepSeek-R1-Distill-Llama-70B**

---

```
Q: Who was in the Hallway when Michael first appeared in the
Garden?
A: Mary
```

---

```
<think>
Okay, so I need to figure out who was in the Hallway when
Michael first appeared in the Garden.  Let me break this down
step by step.

First, I'll look through each step to find when Michael first
shows up in the Garden.  The data is structured from step 1 to
step 64, with each step showing the rooms and the people
inside them.  My goal is to scan these steps sequentially
until I find the earliest occurrence of Michael being in the
Garden.

Starting from step 1, I check each room in every step.  In the
early steps like 1-5, Michael isn't in the Garden yet.  He's
in other places like the Office or Bedroom.  As I go through
each step, I keep an eye on the Garden room.

Continuing this way, at some point, I notice that in step 44,
the Garden has ['Michael'].  That seems to be his first
appearance there because prior steps don't show him in the
Garden.  So, Michael first appears in the Garden at step 44.

Now, once I've identified step 44 as the moment Michael enters
the Garden, I need to check who was present in the Hallway at
that exact step.  Looking back at step 44's data:

{'Kitchen': [], 'Bathroom': [], 'Garden': ['Michael'],
'Office': ['Sandra', 'Daniel'], 'Bedroom': ['John'],
'Hallway': ['Mary']}.

From this, it's clear that in the Hallway during step 44, Mary
is present.  Therefore, when Michael first appeared in the
Garden, Mary was in the Hallway.

I should double-check my findings to ensure accuracy.
Scanning earlier steps confirms that Michael wasn't in the
Garden before step 44, and at step 44, only Mary is in the
Hallway.  No one else is there except her.

So, putting it all together, the person in the Hallway when
Michael first entered the Garden was Mary.
</think>

{"answer": "Mary"}
```

---

Table 3: A generic reasoning example of the largest used DeepSeek model on the FA-RCFA-C task.

---

**DeepSeek-R1-Distill-Llama-70B**

---

```
Q: Who spent the least amount of time alone in the Hallway?
A: John
```

---

```
<think>
Alright, let's try to figure out who spent the least amount of
time alone in the Hallway. So, I have data from step 1 to
step 128, each showing which rooms people are in at that step.
My task is to go through each step, check who's in the
Hallway, and see if they're alone there. Then, count how many
times each person was alone in the Hallway and find out who
had the smallest number.

First, I need to understand what "alone" means. It means that
during a particular step, only that person is present in the
Hallway; nobody else is there. So, for each step, I'll look
at the Hallway entry in the 'rooms' dictionary. If it has
exactly one person, that person was alone that step.
Otherwise, even if multiple people are there, as long as
someone else is present, it doesn't count as being alone.

So, my plan is:

1.  Iterate through each step from 1 to 128.
2.  For each step, get the list of people in the Hallway.

<...>

- Step 1:  Hallway = [Sandra] → Sandra +1 → Sandra=1
- Step 2:  Hallway = [Sandra] → Sandra +1 → Sandra=2
- Step 3:  Hallway = [Sandra] → Sandra=3
- Step 4:  Hallway = [] → No one
- Step 5:  Hallway = [] → No one
- Step 6:  Hallway = [] → No one
- Step 7:  Hallway = [Mary] → Mary=1
- Step 8:  Hallway = [Mary, Michael] → Both present → Not alone
- Step 9: Hallway = [Mary, Michael] → Not alone
- Step 10: Hallway = [] → No one

<...>

So, from this, Michael was alone in the Hallway during step
36, and John was alone during steps 72 and 73. Therefore,
since the question asks for who spent the least amount of time
alone in the Hallway, the answer is Michael.
</think>

{"answer": "Michael"}
```

---

Table 4: Failure case of a reasoning model on a dense context task, DC-SA-C. Although the model correctly outlines a valid general reasoning strategy, it decides to list all observations (orange), making a mistake outlining several steps (red), and ultimately produces the wrong final answer.

