# OpenReview forum: "MMReD: a Cross-Modal Benchmark for Dense Context Reasoning"
_ICLR.cc/2026/Conference — ICLR 2026 Poster_

### Official Review · Reviewer_mQFA · 2025-10-16

**Soundness:** 4
**Presentation:** 3
**Contribution:** 3
**Rating:** 8
**Confidence:** 5

**Summary:**

This paper introduces MMReD (Multi-Modal REasoning in Dense context), a new benchmark designed to evaluate the ability of large language models (LLMs) and large vision-language models (LVLMs) to reason over long, information-rich sequences. The authors argue that existing benchmarks, which primarily use a "Needle-in-a-Haystack" (NIAH) setup, are insufficient as they only test a model's ability to retrieve a specific fact from a large, mostly irrelevant context. MMReD, in contrast, creates scenarios where every piece of the context is important, forcing models to identify and interpret global patterns rather than just locating sparse information.

The benchmark comprises 24 tasks that test models on tracking entities, spatial relationships, and event-based reasoning over sequences of varying lengths, up to 128 observations.
The evaluation revealed a consistent and significant drop in performance across all tested models—including advanced LLMs, LVLMs, and reasoning-specialized architectures—as the sequence length increased.
The study demonstrates that state-of-the-art models fail to generalize to dense context reasoning challenges. Furthermore, standard fine-tuning techniques like Supervised Fine-Tuning (SFT) and GRPO were found to be insufficient for enabling this capability.

The paper concludes that there is an inherent limitation in current model architectures when it comes to reasoning over dense, multi-modal contexts. The MMReD benchmark effectively highlights this gap and underscores the need for new architectural innovations and training paradigms to advance long-context understanding in AI systems.

**Strengths:**

Novel Problem Formulation: The paper introduces "dense context reasoning" as a distinct capability from standard "Needle-in-a-Haystack" (NIAH) retrieval, a significant conceptual shift.
Creative Benchmark Design: MMReD uses a minimalist visual and linguistic environment to effectively isolate reasoning capabilities from perceptual or language complexity.
Clear Writing and Structure: The paper is well-written and logically organized, with an effective abstract and introduction that clearly motivate the work .
Challenges Evaluation Paradigms: The work significantly challenges the sufficiency of the dominant NIAH paradigm, showing it is not a reliable indicator for complex reasoning.

**Weaknesses:**

The paper's central conclusion is that the observed performance degradation reveals an "inherent limitation in current model architectures". While the extensive experiments show that models fail, the paper provides limited insight into why they fail at a mechanistic level.

The claim is a high-level one, and the analysis stops short of diagnosing the specific architectural components that are failing. It is unclear if the bottleneck lies in the attention mechanism's ability to handle uniform information density, the decay of information in positional embeddings over long distances, or how representations are processed through successive layers.

The work could be substantially improved by including more targeted analysis to diagnose the failure modes. An investigation of the models attention patterns on long dense-context (DC) tasks would be highly valuable. Visualizing attention maps could reveal whether attention becomes overly diffuse, incorrectly focuses on recent tokens (recency bias), or fails to integrate information from distant parts of the sequence.

**Questions:**

1. Could the authors comment on this distinction between reasoning "length" and "depth"? Do you believe the architectural limitations observed are primarily related to failures in long-term memory and information aggregation, or do they also impact the ability to construct complex, multi-step logical inferences? How might the models you tested perform on tasks requiring deeper causal or counterfactual reasoning over the same long contexts?
This is not a criticism of the current design but an opportunity to add valuable nuance. A brief discussion on this "length vs. depth" axis of complexity in the conclusion could help frame the current results more precisely and outline a clear roadmap for the next generation of dense context benchmarks, which might incorporate tasks with deeper logical requirements.

2. Could you provide more specific insights into the nature of this limitation? For instance, did your preliminary analysis reveal specific failure modes in the self-attention mechanism, such as attention becoming overly diffuse and uniform on Dense Context (DC) tasks compared to Needle-in-a-Haystack (NIAH) tasks? Or is the failure more related to information decay and representational corruption as information is passed through successive layers?
The paper's impact could be significantly strengthened by including a qualitative analysis, perhaps in the appendix, that delves into these potential causes. For example, providing attention visualizations for a successful short-context DC task versus a failed long-context one could offer powerful visual evidence to support your central claim and diagnose the failure mode more precisely.

---

> ### Author Response · Authors · 2025-11-26
>
> We thank Reviewer mQFA for highlighting that MMReD introduces a significant conceptual shift by framing dense context reasoning; we address the remaining questions below.
>
> **W1 (Q2). Lack of the architectural diagnosis.**
>
> We appreciate this point and agree that a layer-level or attention-pattern analysis could offer additional insight into *why* current architectures struggle. Our goal in this work, however, is to establish and characterize the phenomenon that existing LLM and LVLM architectures consistently fail under dense-context (DC) reasoning, even when perception is simplified, language is controlled, and all extraneous confounds are removed. The benchmark therefore focuses on revealing the limitation.
>
> To ensure that the limitation is not attributable to straightforward fixes, we conducted a broad set of adaptation ablations: larger model sizes, multimodal instruction tuning, coding-oriented tuning, video-centric pretraining, SFT, GRPO, and (as added in the response to Reviewer MPFi) in-context learning. None of these reliably overcome the DC problem. Breadth of these results supports the claim that the bottleneck reflects a core architectural limitation.
>
> We see attention-level analysis as a valuable next step. For NIAH tasks, inspecting attention patterns over the input sequence could reveal whether models (i) incorrectly attend to the wrong step or (ii) diffuse attention across multiple steps. The latter connects directly to uncertainty quantification directions outlined in the end of Sec. 4\. For DC tasks, analyzing attention maps may highlight whether attention becomes uniformly spread across the sequence, hence errors arise from an inherent expressive limitation of the attention function class itself. Such findings would motivate the need for mechanisms that go beyond standard attention and recurrent memory, as discussed at the end of Sec. 4\.
>
> We plan to incorporate this more detailed attention-pattern study in the revision, but consider the development of new mechanisms or solutions out of scope of the present benchmark paper.
>
> **Q1. Reasoning “length” vs. “depth.”**
>
> In MMReD, we explicitly isolate reasoning length from reasoning depth: all benchmark tasks have “zero” depth (no nested or multi-step logical inference), as described in Sec. 3\. Our goal is to probe models’ ability to integrate information over long contexts (length) without confounding effects from multi-step reasoning (depth). Notably, even at this minimal logical complexity, models struggle, revealing that failures arise primarily from limitations in **long-term memory and information aggregation** rather than deep logical construction.
>
> Tasks requiring deeper reasoning would likely exacerbate these difficulties. Measuring such multi-step logical capabilities is an important direction, but it is out of scope for the current benchmark.

---

### Official Review · Reviewer_6qaP · 2025-10-24

**Soundness:** 3
**Presentation:** 3
**Contribution:** 3
**Rating:** 6
**Confidence:** 3

**Summary:**

This paper introduces a new benchmark, MMReD (Multi-Modal Reasoning in Dense context), to evaluate the ability of large LLMs and LVLMs to perform complex reasoning over long sequences of information. The authors argue that existing benchmarks do not adequately test a model's ability to reason in "dense context" scenarios where all information is potentially relevant. MMReD gives models a series of images depicting characters in different rooms and asks questions that require tracking entities, understanding spatial relationships, and identifying global patterns across the entire sequence. The study evaluates a wide range of frontier models and reveals a consistent drop in performance as the context length increases. The results show that even the most advanced models struggle with these dense context tasks. Also, common fine-tuning techniques offer little improvement.

**Strengths:**

* The paper argues that "Needle-in-a-Haystack" tasks (i.e., testing retrieval) are fundamentally different from "dense context" tasks, which require integration and reasoning. Figure 4 provides evidence that performance on these two task types does not correlate well at longer context lengths. This shows that dense context is fundamentally different from retrieval tasks, such as the needle-in-a-haystack problem.

* The authors simplify the visual and linguistic elements and isolate the  reasoning capabilities they want to measure. This avoids confounding variables and ensures that the results reflect its capacity for dense context reasoning. Furthermore, the measure to ensure unique sequences and balanced answer distributions prevents models from relying on memorization or statistical shortcuts, making the evaluation more robust.

* The paper also conducts a series of ablation studies. By testing the effects of model size, fine-tuning methodologies (SFT and GRPO), multimodal adapter types, and video-specific pooling methods, the authors provide multiple views on the problem. The finding that reasoning-specialized LLMs outperform complex LVLMs on the textual version of the tasks is particularly interesting. Specifically, multimodal instruction tuning and video-centric pretraining may be detrimental to this type of reasoning. This suggests that the bottleneck is not multimodal perception but a limitation in long-sequence reasoning.

**Weaknesses:**

* The benchmark's controlled & minimalist design is a potential weakness. The environment is a highly structured, symbolic "toy world" with fixed rules (a 2x3 grid, one character moves per step). This does not reflect the ambiguity and chaotic nature of real-world scenarios, which involve complex simultaneous interactions, occlusions, and less predictable patterns. While an ablation with synthetic "perceptual noise" is included, it doesn't capture the true complexity of real-world visual and narrative understanding.

* Because the benchmark is generated in a way that follows clear logical rules, there is a risk that future models could add an ad-hoc module specifically to solve MMReD-like tasks without truly developing generalized dense context reasoning. For example, a model could develop a specialized internal module for tracking entities in a grid.

* The paper categorizes questions into two main buckets: Needle-in-a-Haystack (NIAH) and Dense Context (DC). However, the degree of "density" is somewhat ambiguous. An analysis of how performance degrades based on the degree of context density required to solve the question could yield deeper insights.

**Questions:**

* The poor performance of video-oriented LVLMs was quite surprising. Do you have any hypotheses for this? Could it be related to their training strategies or a bias in their training data?

* Do you believe the long-context reasoning is an inherent limitation of the Transformer architecture, or is it a problem that can be addressed with different training data and methodologies?

* The paper focuses on tracking and counting tasks. Have you considered expanding the benchmark to include more abstract forms of reasoning, such as inferring intentions or predicting future states based on observed patterns?

---

> ### Author Response · Authors · 2025-11-26
>
> We thank Reviewer 6qaP for highlighting that MMReD provides clear evidence on DC-NIAH fundamental difference; and we address the remaining points of critique below.
>
> **W1. Controlled and minimalist environment**
>
> The controlled design of MMReD is **intentional and central** to its purpose: to isolate DC reasoning without entangling it with real-world confounds such as perception errors, multi-object occlusion, semantic ambiguity, or other complex forms of reasoning. MMReD focuses on measuring the identified reasoning bottleneck itself, using a fully informative sequence where ambiguity and sparsity are removed by design.
>
> To test whether our conclusions depend on the specific instantiation of the environment, we already include an ablation with synthetic occlusion-style perturbations (Sec. 4.2), showing that key conclusions and model rankings remain stable under increased perceptual difficulty. Plus, we evaluated a symbolic variant of the environment (see our response to Reviewer MPFi’s W1) – models performance remain statistically unchanged.
>
> In summary, the benchmark’s controlled design is a deliberate methodological choice; and we argue that provided ablations demonstrate MMReD’s conclusions could be generalized, scaled closer to real-world environments.
>
> **W2. Benchmark-specific shortcut modules**
>
> We agree that any controlled benchmark can, in principle, be overfit by introducing an ad-hoc mechanism tailored to its structure. This is true for similar reasoning benchmarks – from bAbI to NIAH (e.g., similar to detector oracle in Visual Haystacks (Wu et al., 2024)).
>
> MMReD’s contribution is not to mimic the full richness of real-world scenes, but to formalize a clear, reproducible test of dense-context integration, which is currently missing in the literature. Moreover, as we show that the benchmark conclusions are robust to environmental modifications, we can freely vary visual projection in the future, allowing us to test the generalization ability of the models.
>
> **W3. Degree of density**
>
> In MMReD, the NIAH vs. DC distinction is defined by design rather than by a continuous density parameter: NIAH contains a single informative observation embedded in distractors, while DC sequences are fully informative at every step. And our goal is to isolate reasoning over uniformly dense sequences, not to study the full spectrum between sparse and dense contexts. Nevertheless, the benchmark already provides a partial view of density sensitivity:
>
> While most DC tasks have density 1, NIAH tasks have an effective density of 1/N, depending on the number of frames N. Since we vary N, this partially reveals the behavior of increasing density from sparse starting points. Notably, the sharp decline in quality and DC-NIAH divergence emerge around N=32, giving an estimate of the minimum density gap at which DC-like effects begin to manifest.
> As seen qualitatively in Figures 8–13, the easiest DC task is “DC-CR-R.” Quantitatively, models score 54.6% accuracy on this task versus 32.9% on other DC tasks on average. This question involves identifying the room crowded for the most steps (a pattern that appears every second step on average and is arguably the easiest to detect), effectively reducing the density by a factor of two and partially offsetting performance decline – pattern observed in Figures 8–13.
>
> Thus, while a continuous density parameter is not explicitly defined, MMReD already exposes models to varying densities; resulting behavior is predictably monotonous. We will clarify this in the text and note that fully parametric density variants could be explored in future work.
>
> **Q1. Poor performance of video-oriented LVLMs**
>
> Our main hypothesis is that instruction tuning biases LVLMs toward narrative, semantic, and locally salient cues rather than the precise symbolic tracking required by MMReD. This is consistent with our observation (Sec. 4.2, lines 371–373) about LVLM-LLM gap, likely due to (i) token-budget shifts toward visual inputs, reducing the effective capacity for reasoning, and (ii) catastrophic forgetting from vision-domain finetuning.
>
> **Q2. Long-context limitation**
>
> The failure persists across model sizes, instruction tuning, coding tuning, video pretraining, SFT, GRPO, and ICL, indicating that it is not resolved by existing training methodologies or datasets. As noted in Sec. 4, overcoming this likely requires architectural or algorithmic innovations beyond standard attention and memory mechanisms.
>
> **Q3. Expanding to abstract forms of reasoning**
>
> Our benchmark is intentionally restricted to fully observable, fact-based reasoning so that DC integration can be measured without confounds such as learning dynamics or policy prediction. Besides, MMReD already includes a structured internal model (only one entity can move per step), and models can in principle learn this, but observably fail (Tab. 4). Such extensions are interesting future directions, but not part of our goal of isolating DC reasoning.

---

### Official Review · Reviewer_MPFi · 2025-10-25

**Soundness:** 3
**Presentation:** 3
**Contribution:** 2
**Rating:** 6
**Confidence:** 4

**Summary:**

The paper introduces a new multimodal (vision + text) benchmark for dense context reasoning. Compared to existing benchmarks, it challenges models to identify and interpret global patterns across entire contexts. The tasks vary in difficulty, with a good number being challenging for existing models. Investigation also shows conventional fine-tuning techniques do not help improve the performance and so more advanced strategies will be needed to address the challenge.

**Strengths:**

* The benchmark is well motivated and studies a relatively valuable point about dense context reasoning in multimodal tasks.
* A large number of models is evaluated, including many recent ones.
* The benchmark has setups that are doable for existing models, but also setups that are too challenging for current models, so the benchmark is likely to be relevant for at least some time.
* Various related questions are studied, and these help probe the performance of the models on the task further.
* The paper is clearly written and easy to read.

**Weaknesses:**

* All tasks come from one environment, so one could say that the benchmark is somewhat limited in this sense. A benchmark with e.g. three more-distinct types of setups would enable more rigorous investigation of dense context reasoning.
* There could be a short discussion where the performances of models for text-only vs multimodal versions of the benchmark are compared. Which one is easier / to what extent is the text-only version easier? (one can infer this from the tables though)
* In-context learning may potentially help so could be interesting to investigate too given its popularity.
* Minor: the plots would benefit from larger font sizes where possible.

**Questions:**

* How do the performances of models for text-only vs multimodal versions of the benchmark compare?
* Does in-context learning help for these sorts of tasks?

---

> ### Author Response · Authors · 2025-11-26
> **Methodology and Environment Design**
>
> We thank Reviewer MPFi for highlighting that MMReD addresses a valuable and under-explored aspect of multimodal reasoning, the integration of global patterns across dense contexts. Below we clarify several points raised in the critique.
>
> **W1. Use of a single environment.**
>
> Our use of a single controlled environment is deliberate and grounded in the core design goals of MMReD. The benchmark is not tied to a specific “room-character” aesthetic but is a projection of a general theoretical framework: reasoning over a temporally evolving set of entities and relations (formalizable as a dynamic graph or cellular automaton). In this sense, the current environment is simply one projection of this framework; its representation (number of entities, interaction types, spatial layout, symbolic vs. visual encoding) could be systematically varied without altering the underlying Dense Context (DC) reasoning problem.
>
> While extending MMReD to multiple distinct environments is conceptually straightforward, it entails a linear increase in computational cost regarding the number of projections. Given our strong theoretical expectation that the core reasoning challenges would persist, we prioritized depth of analysis over multiplying the environmental footprint. To address the reviewer’s concern regarding robustness, we provide two key pieces of evidence:
>
> 1. **Visual Robustness (Sec. 4.2):** We introduced systematic perturbations to the visual context (controlled occlusion noise). Despite these alterations to the perceptual structure, all key conclusions and model rankings remained stable, demonstrating that MMReD’s signal is robust to meaningful modifications of the visual input.
> 2. **Symbolic Environment Ablation (New):** We evaluated a subset of LLMs on a symbolically encoded variant of the environment using a 5-location $\\times$ 6-entity setup (L1–L5 and E1–E6). This configuration preserves the combinatorial complexity while providing a structurally distinct projection (no grid geometry, altered factorization, and abstract symbolic naming). As expected, model rankings and NIAH-DC gaps remained statistically unchanged compared to the standard environment.
>
> These observations suggest that the benchmark’s conclusions arise from the underlying DC reasoning design rather than superficial aspects of the scene. We will include this symbolic ablation in the final revision to further validate the benchmark's generalizability.

---

> > ### Comment · Reviewer_MPFi · 2025-11-27
> > **Response**
> >
> > Thank you very much for the additional explanations and experiments. I find these valuable and useful for investigating the contribution in more depth. I wonder if the results for Symbolic Environment Ablation are reported somewhere as I did not manage to find them.

---

> > > ### Author Response · Authors · 2025-12-03
> > > **Symbolic Environment Ablation**
> > >
> > > We thank the reviewer for the positive assessment and for highlighting the importance of verifying environment robustness. We confirm that the **Symbolic Environment Ablation** results presented below were conducted specifically during the rebuttal phase to address this concern. These results, along with the detailed breakdown, will be included in the **Appendix** of the camera-ready revision to demonstrate the generalizability of MMReD across modalities and environmental semantics.
> > >
> > > ### **Correlation of performance in different environments**
> > > Table 3 reports the Pearson correlation coefficient $r$ between model performance in the standard environment (visual/textual "rooms & characters") and the introduced abstract symbolic variant across varying sequence lengths $N$.
> > >
> > > **Table 3: Pearson Correlation of performance in Standard vs. Symbolic environments.**
> > >
> > > | Sequence Length ($N$) | 1 | 2 | 4 | 8 | 16 | 32 | 64 | 128 |
> > > | :--- | :---: | :---: | :---: | :---: | :---: | :---: | :---: | :---: |
> > > | **Pearson Correlation** | $0.98 \pm 0.09$ | $0.95 \pm 0.16$ | $0.80 \pm 0.30$ | $0.72 \pm 0.35$ | $0.69 \pm 0.36$ | $0.89 \pm 0.23$ | $0.86 \pm 0.25$ | $0.83 \pm 0.28$ |
> > >
> > > We observe a consistently high correlation ($r > 0.7$) across all context lengths, peaking at $r > 0.98$ for short contexts. This strongly suggests that model rankings are consistent regardless of the specific environmental projection, validating that MMReD measures a fundamental **Dense Context Reasoning** capability rather than overfitting to surface-level features.
> > >
> > > ### **Performance Gap Analysis (Symbolic - Original)**
> > > Table 4 details the relative performance impact of shifting to the symbolic domain, calculated as $\Delta = \frac{\text{Symbolic} - \text{Original}}{\text{Original}}$ (in %).
> > >
> > > **Table 4: Relative Performance Improvement ($\Delta$) in Symbolic Environment.**
> > >
> > > | Model | $N=1$ | $N=2$ | $N=4$ | $N=8$ | $N=16$ | $N=32$ | $N=64$ | $N=128$ |
> > > | :--- | :---: | :---: | :---: | :---: | :---: | :---: | :---: | :---: |
> > > | **Qwen3-30B-A3B** | $-16.0$ | $-6.3$ | $-1.3$ | $0.9$ | $-6.7$ | $-5.7$ | $-5.0$ | $-3.5$ |
> > > | **Qwen3-Next-80B-A3B** | $-18.8$ | $-4.0$ | $1.4$ | $4.2$ | $-4.8$ | $-7.0$ | $3.0$ | $2.7$ |
> > > | **Qwen3-4B** | $-12.3$ | $5.9$ | $8.1$ | $9.7$ | $8.7$ | $-2.6$ | $-5.1$ | $7.1$ |
> > > | **Qwen3-8B** | $-6.9$ | $-5.9$ | $-10.1$ | $-6.8$ | $-21.3$ | $-10.5$ | $-7.9$ | $9.6$ |
> > > | **Qwen3-14B** | $-15.5$ | $-3.2$ | $3.8$ | $10.3$ | $12.0$ | $8.9$ | $6.3$ | $18.3$ |
> > > | **Qwen3-32B** | $-16.2$ | $-11.6$ | $-14.0$ | $-13.0$ | $-17.1$ | $-9.3$ | $-13.4$ | $-4.6$ |
> > > | Mean $\pm$ Std| **$-14.3 \pm 4.2$** | **$-4.2 \pm 5.9$** | **$-2.0 \pm 8.3$** | **$0.9 \pm 9.2$** | **$-4.9 \pm 13.4$** | **$-4.4 \pm 7.1$** | **$-3.7 \pm 7.4$** | **$4.9 \pm 8.6$** |
> > >
> > > ### **Discussion**
> > > The results reveal a distinct dichotomy between initial grounding and long-context tracking:
> > >
> > > * **The "Semantic Grounding" Effect ($N=1$):** We observe a sharp performance drop (Mean $\Delta = -14.3\%$) at sequence length 1. This indicates that natural language semantics (e.g., "Kitchen") significantly aid LLMs in initial state grounding compared to abstract tokens. This aligns with recent findings by Wang et al. (2025), who demonstrate that LLMs struggle with pure symbolic representations lacking natural language semantics.
> > > * **Convergence at Dense Contexts ($N > 1$):** As sequence length increases, the performance gap narrows significantly. At extreme lengths ($N=128$), the mean gap shifts to positive ($+4.9\%$). This suggests that for dense context reasoning, the difficulty stems from the structural complexity of state tracking rather than semantic interpretation. In certain cases (e.g., Qwen3-14B), abstract symbols may even improve performance by reducing distraction from semantic priors, allowing the model to focus purely on state transitions.
> > >
> > > **References**
> > >
> > > *Wang et al. (2025). Leveraging Language-based Representations for Better Solving Symbol-related Problems with Large Language Models. COLING.* https://aclanthology.org/2025.coling-main.372

---

> ### Author Response · Authors · 2025-11-26
> **Experimental Analysis and Ablations**
>
> **W2 (Q1). Multimodal vs. Text-only performance.**
>
> To address the comparison between text-only and multimodal performance, we computed the relative performance gap between the Text-only (LLM) and Multimodal (LVLM) versions of the evaluated models. The metric is defined as $\\frac{\\text{LLM} \- \\text{LVLM}}{\\text{LLM}}$, where a positive value indicates the Text-only model outperforms the Multimodal version.
>
> Table 1: Relative Performance Gap ($\\%$) of Text-only vs. Multimodal Models
>
> | Model | N=1 | N=2 | N=4 | N=8 | N=16 | N=32 | N=64 | N=128 |
> | :--- | :--- | :--- | :--- | :--- | :--- | :--- | :--- | :--- |
> | Qwen2.5-3B-Instruct | 6.7 | 4.1 | \-2.3 | \-5.7 | \-12.0 | \-23.2 | \-32.2 | \-37.5 |
> | Qwen2.5-7B-Instruct | 2.3 | \-4.0 | \-11.3 | \-9.1 | \-1.8 | \-9.6 | \-3.9 | \-19.1 |
> | Qwen2.5-72B-Instruct | 0.6 | 1.2 | 0.6 | 2.9 | 5.7 | 3.9 | 6.2 | \-3.8 |
> | GPT-4o | 9.8 | 7.1 | 8.3 | 6.0 | 7.2 | 3.8 | \- | \- |
> | GPT-4o-mini | 7.2 | 12.6 | 14.8 | 14.7 | 21.0 | 27.6 | 32.5 | 27.6 |
> | Mean | 11.0 | 16.8 | 35.1 | 35.3 | 34.1 | 12.5 | 0.7 | \-8.2 |
>
> *Note: Positive percentages indicate the Text-only model performed better.*
>
> Text-only models generally outperform their multimodal counterparts (Mean gap $\\approx 35\\%$ in mid-contexts), likely because textual descriptions abstract away visual perception costs. However, this trend depends heavily on model scale. Large models (e.g., GPT-4o, Qwen-72B) maintain a consistent but small advantage for text ($0-10\\%$). Conversely, smaller models (e.g., Qwen-3B/7B) exhibit an inversion at long contexts ($N \\ge 64$), where multimodal performance becomes superior. This suggests that for limited-capacity models, processing the massive token load of detailed text descriptions is more taxing than handling compressed representations from a vision encoder.
>
> **W3 (Q2). In-context learning.**
>
> We agree that In-Context Learning (ICL) is a relevant direction. We performed an ablation where we provided each model with five solved examples (from a separate set, up to length 16\) appended to the prompt.
>
> Table 2: Relative Performance Improvement via In-Context Learning ($\\%$)
>
> | Sequence Length | N=1 | N=2 | N=4 | N=8 | N=16 | N=32 | N=64 | N=128 |
> | :--- | :--- | :--- | :--- | :--- | :--- | :--- | :--- | :--- |
> | Mean Improvement | 13.4 | 16.9 | 17.9 | 17.2 | 13.8 | 7.5 | 7.7 | 12.5 |
>
> As expected, ICL improves performance: across all tested LLMs (4B to 32B), we observe an average $+13.8\\%$ relative gain. For mid-range models, the effect is stronger ($+23.1\\%$). However, even with these gains, performance remains far below a "passing grade" for DC-type tasks.
>
> Furthermore, when compared to fine-tuning approaches on the same mid-range models, SFT yields $+57.7\\%$ and GRPO yields $+27.8\\%$, both surpassing ICL. As discussed in Sec. 4.3, even SFT and GRPO fail to resolve the core DC reasoning limitations. Thus, while ICL is helpful, it is fundamentally insufficient to solve the dense context problem. We will include these results in the revised manuscript.
>
> **Response to W4. Plot font sizes.**
>
> We appreciate this suggestion. We will increase the font sizes in the final revision, specifically for Figures 2 and 6, to ensure optimal readability.

---

### Official Review · Reviewer_URmZ · 2025-10-28

**Soundness:** 4
**Presentation:** 3
**Contribution:** 3
**Rating:** 6
**Confidence:** 3

**Summary:**

The paper presents MMReD, a benchmark for evaluating long-context and cross-modal reasoning in large language and vision-language models (LLMs and LVLMs). Unlike traditional “Needle-in-a-Haystack” (NIAH) setups focused on sparse retrieval, MMReD emphasizes dense context reasoning, where all observations are informative and models must integrate global patterns rather than locating single facts.

**Strengths:**

1. MMReD effectively bridges the gap between retrieval-based and integrative reasoning benchmarks by introducing dense, information-rich multi-modal contexts that better reflect real-world reasoning challenges.
2. The evaluation suite is extensive and well-designed, covering a wide range of strong baseline and state-of-the-art models across relevant modalities and tasks.
3. Comprehensive ablation studies offer deep insights into model behavior, highlighting the limitations of current approaches in handling dense contextual reasoning.

**Weaknesses:**

1. The benchmark is constructed in an artificial and controlled environment, which may limit the model's ability to generalize the the real world modalities and with more noises.
2. The close-ended question with exact-match accuracy may limit the insight into different models' performance in dense context reasoning.

**Questions:**

1. How do you envision MMReD transferring to real-world multimodal reasoning, such as video QA? Could introducing controlled perceptual noise or natural imagery affect the benchmark’s conclusions?
2. Since all contexts are synthetic, do you plan to integrate text–image mixtures or human-annotated long-context tasks to bridge synthetic and natural settings?

---

> ### Author Response · Authors · 2025-11-26
>
> We thank Reviewer URmZ for recognizing the novelty of MMReD in bridging retrieval-based and integrative reasoning benchmarks; below we clarify our position on the identified weaknesses and address the raised questions.
>
> **W1. Artificial and controlled environment.**
>
> The synthetic controlled setup is an **intentional and crucial design choice** (Sec. 3.1, lines 148--155): MMReD isolates dense-context (DC) reasoning without confounds from visual complexity, OCR errors, linguistic ambiguity, and logical depth. This allows us to directly measure the capability we target — uniform integration of densely informative long contexts, which existing benchmarks do not test.
>
> Importantly, we explicitly evaluated generalization under perceptual noise. Sec. 4.2 introduces synthetic occlusion-like perturbations (5% error rate) and shows that while absolute accuracy decreases proportionally, all key trends and conclusions (e.g., NIAH vs. DC, Fig. 5\) remain unchanged. Thus, we show that MMReD’s evaluation is stable under realistic perturbations so its conclusions could be generalized to real-world scenarios.
>
> We will clarify this motivation and design-principles in the revision.
>
> **W2. Close-ended questions.**
>
> Exact-match answers to such questions ensure that performance reflects reasoning rather than natural-language generation. And we intentionally avoid multi-step symbolic chains so that we can isolate DC reasoning difficulty rather than confound it with depth of logical inference.
>
> Despite being syntactically simple, the 24 question types span object tracking, temporal and spatial reasoning, counting, and global statistics, covering core reasoning categories and presenting a diverse evaluation (Sec. 3.3). Their difficulty is evident in the empirical results: several strong models collapse to 0% accuracy at N \= 128, indicating that the formulation is sufficiently challenging while remaining interpretable and controlled.
>
> We will highlight this design rationale more clearly in the revised manuscript.
>
> **Q1. Transfer to real-world multimodal reasoning.**
>
> DC reasoning – tracking multiple entities over long horizons, aggregating global patterns – is a core component of real-world video QA. Our ablation (Sec. 4.2) shows that adding controlled perceptual noise does not alter the benchmark’s conclusions, indicating that MMReD captures a reasoning limitation that persists under more realistic conditions. Hence, MMReD is designed as a diagnostic component that can complement natural-video tasks.
>
> **Q2. Integration of natural settings.**
>
> Synthetic sequences allow precise control over structure and representation of tasks and guarantee that we measure DC reasoning directly. Extending MMReD with natural images or human-annotated long-context data is a promising **future direction**, but intentionally out-of-scope for this benchmark, whose contribution is isolating the DC reasoning component.
>
> We thank the reviewer again and will clarify these points in the revision, specifically expanding discussion of real-world applications and MMReD design principles.

---

### Author Response · Authors · 2025-12-03

We thank all reviewers for careful reading and constructive feedback; their critiques helped us sharpen the paper’s message and motivated several clarifications and new experiments resulting in valuable insights. Below, we summarize our key responses that will be included in the revision.


Additions:
1. **Symbolic-environment ablation.** To address environment-dependence concerns we added a symbolic projection (L1–L5, E1–E6) and evaluated several LLMs. Model rankings and overall trends remain stable. We will include full tables and correlations in the Appendix.
2. **In-context learning ablation.** We added a controlled ICL experiment (solved examples for length-16 tasks) and reported average gains: +13.8% across LLMs and +23.1% for mid-range models. These gains are real but modest compared to SFT (+57.7%) and GRPO (+27.8%) on the same mid-range models, and, critically, none of these methods fully solve the DC problem.
3. **Readability fixes.**


Clarifications about methodology and scope:
1. **Why a controlled, minimalist environment?** It is an intentional design choice to isolate the DC reasoning capability by removing perception and linguistic confounders, making failures interpretable and reproducible. We explicitly tested robustness to perceptual noise (Sec. 4.2) and to structural re-projections (symbolic ablation) – both preserve our central conclusions – justifying a possible extension toward natural images and human annotations.
2. **Why exact-match, close-ended questions?** Exact-match answers remove evaluation noise from generation/formatting and let us attribute errors to reasoning integration rather than verbalization. Also, designed tasks already produce very challenging behavior (several strong models hit 0% at N=128), preserving the benchmark both controlled and demanding.
3. **Single-environment and shortcut risk.** We acknowledge that our benchmark (as most others) can be shortcuted; however the symbolic and noise ablations indicate our findings are not surface-specific and generalize robustly beyond selected representation.
4. **Why models fail?** For LVLMs and video-oriented LVLMs underperformance, our working hypothesis is that token-budget shift toward visual inputs biases models toward narrative local cues rather than precise symbolic tracking, also reducing reasoning token budget, plus catastrophic forgetting from vision-domain fine-tuning possibly affects performance. This is consistent with our explicit observation that LLMs outperform their LVLM counterparts (added in the response). We also agree that mechanistic analysis, such as attention maps, would reveal valuable insights; meanwhile, presented extensive negative results (scale, reasoning, SFT/GRPO, ICL, video pretraining, etc.) already indicate that the failure is not trivially fixable by the existing approaches and architectures.


We are grateful for the reviewers’ suggestions and have used them to (i) add robustness experiments (symbolic ablation, ICL), (ii) improve presentation, and (iii) plan follow-ups (attention analysis, formal problem definition, in-depth reasoning scaling) which will appear in the revised manuscript. We believe these additions strengthen the paper’s claims while keeping its original goal clear: to reveal a reproducible, cross-modal limitation in current models and to motivate future architectural and methodological research.


Sincerely yours,

Authors

---

### Meta-Review · Area_Chair_8FdX · 2026-01-07

**Summary:**

Across reviews, the submission is viewed as a timely benchmark contribution that cleanly distinguishes dense-context integration from retrieval-style “needle” setups, supported by broad model coverage and probing analyses. The main concerns driving hesitancy are: (i) the controlled/synthetic design and reliance on a single environment (generalization to real-world multimodal settings), (ii) the use of close-ended exact-match scoring (limited behavioral insight), and (iii) limited mechanistic diagnosis behind the headline claim that current architectures exhibit a fundamental limitation. After the rebuttal, the authors added/outlined additional robustness and analysis (symbolic-environment ablation, controlled ICL, explicit text-only vs multimodal comparisons, and clarification of design choices), which strengthens the empirical story and addresses several “completeness” concerns, but the deeper mechanistic explanation remains largely future work. Overall, the net balance supports a borderline-but-positive poster acceptance.

**Reviewer Concerns:**

Addressed concerns:
- Authors justify the controlled design, provide/describe perceptual-noise robustness and a symbolic-environment ablation showing stable trends/rankings; this directly targets “environment dependence” and partially mitigates generalization worries.
- uthors add an ICL ablation and contextualize its gains vs SFT/GRPO, clarifying that ICL helps but does not solve the core problem.

Remaining concerns:
- Robustness checks help, but the extension to natural imagery / human-annotated long-context tasks is explicitly deferred; concerns are softened but not eliminated.
- Authors clarify the benchmark’s categorical framing (NIAH vs DC) and offer partial discussion, but the reviewer’s request for a more continuous density analysis remains only partially satisfied.

**Reviewer Scores:**

- URmZ (6 → 6): rebuttal addresses the stated weaknesses (controlled environment; exact-match) with clearer rationale + robustness, but the reviewer already framed this as a borderline 6 and might remain at 6.
- MPFi (6 → 7): the rebuttal directly adds what they asked for (text vs multimodal comparison; ICL; stronger environment-robustness argument). Given their appreciative follow-up, a +1 seems plausible.
- 6qaP (6 → 6): authors provide hypotheses and clarifications, but real-world complexity/shortcut concerns and density-continuum requests are only partially addressed; likely unchanged.
- mQFA (8 → 8): already supportive; rebuttal clarifies length-vs-depth and scope, but mechanistic diagnosis is still pending; likely unchanged.

---

### Decision · Program_Chairs · 2026-01-26

Accept (Poster)